# Phenotypic Heterogeneity, Bidirectionality, Universal Cues, Plasticity, Mechanics, and the Tumor Microenvironment Drive Cancer Metastasis

**DOI:** 10.3390/biom14020184

**Published:** 2024-02-03

**Authors:** Claudia Tanja Mierke

**Affiliations:** Faculty of Physics and Earth System Science, Peter Debye Institute of Soft Matter Physics, Biological Physics Division, Leipzig University, 04103 Leipzig, Germany; claudia.mierke@uni-leipzig.de

**Keywords:** transmigration, intravascular cancer cell dissemination, tumor associated macrophages (TAMs), exosomes, stiffness, mechanotransduction, cancerogenesis, cancer stem cells (CSCs), fibroblasts

## Abstract

Tumor diseases become a huge problem when they embark on a path that advances to malignancy, such as the process of metastasis. Cancer metastasis has been thoroughly investigated from a biological perspective in the past, whereas it has still been less explored from a physical perspective. Until now, the intraluminal pathway of cancer metastasis has received the most attention, while the interaction of cancer cells with macrophages has received little attention. Apart from the biochemical characteristics, tumor treatments also rely on the tumor microenvironment, which is recognized to be immunosuppressive and, as has recently been found, mechanically stimulates cancer cells and thus alters their functions. The review article highlights the interaction of cancer cells with other cells in the vascular metastatic route and discusses the impact of this intercellular interplay on the mechanical characteristics and subsequently on the functionality of cancer cells. For instance, macrophages can guide cancer cells on their intravascular route of cancer metastasis, whereby they can help to circumvent the adverse conditions within blood or lymphatic vessels. Macrophages induce microchannel tunneling that can possibly avoid mechanical forces during extra- and intravasation and reduce the forces within the vascular lumen due to vascular flow. The review article highlights the vascular route of cancer metastasis and discusses the key players in this traditional route. Moreover, the effects of flows during the process of metastasis are presented, and the effects of the microenvironment, such as mechanical influences, are characterized. Finally, the increased knowledge of cancer metastasis opens up new perspectives for cancer treatment.

## 1. Introduction

Solid primary tumors are not homogeneous accumulations of cells. Instead, they tend to be heterogeneous and consist of different cell types. The heterogeneity of cancers may be due to multiple genetic alterations that develop cancer. This represents a central challenge in cancer biology: to map and subsequently comprehend what kind of somatic genomic alterations cause cancer. The question arises: in the era of cancer genomics, why are scientists in need of comprehending the oncogenicity of cancer driver genes and mutations? Despite the fact that alternative definitions and methods of detection have been established to identify cancer-promoting genes and mutations, thousands of cancer genomes have been examined, resulting in a strikingly convergent collection of approximately 300 genes that are mutated in a minimum of one type of cancer. Nevertheless, many characteristics of these genes and their function in cancer are at present ambiguous, especially when a somatic mutation is genuinely oncogenic. A limited group of driver mutations lead to cancer. Identifying driver mutations poses a major hurdle, both experimentally and computationally, particularly for infrequent drivers [1].

Since it is still difficult to decide whether a primary tumor will progress to a malignant tumor through the development of metastases, there is still a demand for further novel markers, such as physical markers, to identify cancer cells and their malignant development. There is a prevailing concept of the universality of the mechanical phenotype of cancer cells. The remaining important questions are whether the mechanical phenotype is independent of the cancer type and the cancer state. Another question arises: whether the tumor microenvironment (TME) can alter the mechanical phenotype or not? The constitution, architecture, and organization of the extracellular matrix (ECM) are also important elements in the overall view of mechanobiology, as they shape the mechanical characteristics of the ECM and thus determine aspects of the mechanical signals perceived by the cells. However, it has been identified that a bidirectional interaction exists between the mechanosensing of the ECM and restructuring by cells [2,3]. Integrins have been thoroughly investigated as bidirectional connections that link the ECM and the cytoskeleton of the cells [4,5,6]. The bidirectional nature of the interplay between cancer cells and their microenvironment, which includes acellular and cellular elements, is critical for the malignant progression of cancer [7,8]. Cancer cells communicate with endothelial cells or entire vessels to migrate and invade the tissues for metastasis. The interplay between cancer cells and the endothelial cell lining of blood or lymphatic vessels has been explored, but the mechanisms are not yet fully understood. Thereby, the role of macrophages is controversially discussed. However, the aim of this review is to emphasize the role of tumor heterogeneity, universal cues, mechanics, and the microenvironment in cancer metastasis. This review concentrates in particular on the metastatic cascade and the interaction between cancer cells and macrophages. The emphasis is on mechanical factors that favor or hinder the metastasis of cancer cells. Finally, the possibility of cancer cells switching between the different cell shapes during metastasis is also discussed.

## 2. Cancer Development

Cancer can be seen as a collection of diseases that evolve when the processes that regulate normal cell growth, cell division, and the entire life span become dysregulated. Consequently, the cells begin to proliferate out of control, prevent them from dying when they normally ought to, and cause the motility of other cells and tissues like blood vessels, immune cells, and various other types of normal cells to fuel the tumor’s growth advantage relative to the adjacent tissue. When the cancer progresses, specific cells inside the cancer take on specific modifications that offer them and their offspring cells the highest likelihood of growing and surviving. These alterations may involve the capacity to proliferate more rapidly, to manage to persist in the face of medical intervention, to infiltrate neighboring tissues and organs, to escape the body’s immune defense system, to enter the bloodstream and/or lymphatic system, to propagate without entering vascular systems, and to disseminate to remote areas of the organism. The majority of more advanced cancers display some, perhaps all, of these characteristics.

### 2.1. Driver Mutations Advance Cancerogenesis

Cancer leads to an imbalance of fundamental processes in the cell that subsequently cause alterations in DNA and in DNA-repair enzymes. As a consequence, cancer cells acquire multiple (epi)genetic changes in the course of their lifetime, and these changes occur one or two orders of magnitude more often compared to reproductive and normal somatic cells [9]. Consequently, cancer cells collect numerous genetic changes during their lifespan, but only a limited number of them accelerate the progression of the cancer, the so-called driver mutations. Cancerogenesis has long been revealed as a multistage disease. Driver mutations can differ according to the cancer type and the individual patient, and their impact on cancer may be time-delayed or immediate. For example, they can stay dormant for quite some time and turn into drivers only at specific phases of cancer, or they can promote oncogenesis only in combination with additional mutations. The large mutational, biochemical, and histological heterogeneity of tumors renders the identification of driver mutations very difficult. One of the first findings was that the mortality rate for certain types of cancer increases with the sixth power of the patient’s age. A numerical model has been developed that predicts a number of successive driver mutations and different cancer stages [10]. Additional investigations identified a limited number of mutations that accelerate the development of cancer (driver mutations) [11,12]. For example, roughly a single driver mutation for each patient was detected in sarcoma, thyroid, and testicular cancer, and approximately four driver mutations for each patient with bladder, colorectal, and endometrial cancer [13]. Nevertheless, it is generally agreed that most mutations in cancer are broadly neutral, which are referred to as passenger mutations, and do not encourage the pathogenesis of cancer. The overwhelming majority of driver mutations are either replacements of single nucleotides or point mutations.

Apart from genomic mutations, epigenetics concerns changes in DNA that are not associated with a modification of the DNA base sequence. The epigenetically altered genome is termed the “epigenome”. An important epigenetic process involves the packaging of a cell’s DNA within a cell nucleus. The latest findings strengthen the idea that mutations in the epigenetic signal transduction apparatus, comprising histones, such as K27M mutations in histone H3 in gliomas [14], and chromatin remodelers, such as in the genes ARID1A and ARID1B [15], are prospective new epigenetic markers in cancer [16]. 

### 2.2. Establishment of Basic Primary Cancer Models

Cancer is evoked by a broad range of factors, including chemical carcinogenesis, epigenetic changes, somatic mutations, and viral infections [17,18]. The way normal cells turn into cancer cells is currently based on two different models (Figure 1). Supporters of the deterministic model propose the hypothesis that somatic stem cell transformation by somatic mutations produces a specific subset of cancer cells capable of self-renewal, which are known as cancer stem cells (CSCs) [19,20]. CSCs generate subsidiary cells that have a restricted tumorigenic and metastatic capacity and subsequently develop primary tumors [21,22]. 

The model of clonal evolution of carcinogenesis, which is commonly referred to as the stochastic model, implies that mutations or epigenetic changes bestow selective reproductive predominance on a cell over healthy cells, resulting in unrestrained growth and a primary solid tumor [23,24]. The clonal evolution model is similar to a Darwinian natural selection model, which is an adaptive system. The model of clonal evolution predicts that selection at various phases of tumor growth leads to enhanced genetic and epigenetic changes and a reduction in tumor suppressive defense mechanisms, resulting in susceptibility to oncogenesis [25]. None of these models accounts for the high level of heterogeneity present in primary solid tumors [26]. Recently, a fusion of these two models, referred to as the cellular plasticity model, has been advocated (Figure 1). It assumes that the “original cell” in this case is different from a CSC. Rather, the cellular plasticity model assumes that healthy cells are naturally plastic and may exhibit phenotypic alterations upon being subjected to a stimulus, either internally or externally [27]. This congenital plasticity allows healthy cells to evolve by undergoing epigenetic and phenotypic modifications, enabling transformation into CSCs. Exterior impulses are capable of causing the buildup of multiple mutations throughout the cancer cells, leading to a huge heterogeneity within primary solid tumors [17]. A heterogeneous solid tumor arises from both kinds of mutated cells [28].

Why does phenotypic heterogeneity arise on a mechanistic basis? The expression of genes constitutes a stochastic or “noisy” biological process. This noise can arise in two different types. It can be broken down broadly into two principal categories: Firstly, isogenic cells/individuals that obtain the same pieces of information from the surrounding environment can result in varying expressions of a phenotypic characteristic. Isogenic cells vary because of the noise generated through the process of splitting the cell components binomially during the moment of cell division [29,30]. Consequently, the inherent stochasticity of biochemical processes like transcription and translation produces “intrinsic” noise. Secondly, isogenic cells/individuals that obtain varying pieces of information from the surrounding environment can result in varying expressions of a phenotypic characteristic. Fluctuations in the abundance or states of other cellular compounds elicited by the surrounding environment can indirectly result in fluctuations in the expression of a certain gene and therefore generate “extrinsic” noise [31] (Figure 2).

Does a universal mechanical phenotype of cancer cells exist in general or at specific time points of tumorigenesis and/or malignant progression? The findings of the physical oncology research community point towards the fact that the relationship between the biophysical TME and genetic modification exerts a considerable influence on the progression of tumors. Cancer cells in particular and the stromal cells connected to them modify their individual cytoskeletal and physical characteristics while also restructuring the surrounding microenvironment with abnormal mechanical characteristics [5]. Finally, these altered mechano-omics of cancer tissues and their components profoundly displace the mechanotransduction paradigms in cancer and stromal cells and activate oncogenic cues inside the neoplastic niche to promote cancer progression.

Although there are many possibilities for tumor cell diversity, is there still a possibility that a certain cell phenotype, such as the mechanophenotype, is not different but the same? Moreover, is this mechanophenotype found universally in the different tumors, does it also exist during malignant progression, or is it at least also universally changed? 

## 3. Universal Properties of Cancer and Their Effect on Cancer Cell Function

The mechanical phenotype can be characterized by considering the cell as a material. It is a pivotal factor as it dictates the interaction between forces such as tension, pressure, and entrapment and the consequent alterations in cell morphology, such as cell shape and physical size. The mechanical phenotype can be expressed by various measurements like viscoelasticity, cell deformability, cell adhesion characteristics, and cell shape. To address the question of whether there is a universal mechanical phenotype of cells, it is possible to consider the function of certain proteins, such as Caveolin1 (Cav1). Specifically, the focus is on emphasizing the involvement of caveolae and caveolar constituents, particularly Cav1, in the process of incorporating the mechanical forces transferred by the ECM [32] and the specific involvement of cellular cytoskeletal remodulation as a mechanism of action whereby caveolae and caveolins can react to and remodel the ECM [33,34,35].

In general, it is the cellular mechanical phenotype that determines survival after deformations caused by entrapment and liquid streaming. The concept involves the assumption that cancer cells are sculptural (plastic) and acquire various mechanical phenotypes when exposed to various geometries, which promote their survival. Therefore, an attractive objective of physical cancer research is to interfere with the capacity of cancer cells to take on different mechanical states. For this reason, the mechanosensory process of cancer cells must be investigated. The ability of cells to recognize stiffness is necessary to react to the rigidity of the matrix. It was investigated how healthy cells and cancer cells exhibit discrete mechanical characteristics [36]. Cancer cells appeared softer compared to their healthy control equivalents, irrespective of the specific cancer type, such as breast, bladder, cervix, ovarian, pancreatic, or Ha-RasV12-transformed cells [37]. The measurements were conducted with atomic force microscopy (AFM). When growing on ECM matrices of different stiffness, low stiffness can impair the proliferation of healthy cells [38], whereas this effect is not attained by cancer cells and transformed cells. Hence, cancer cells experience a modification of their mechanical phenotype, which entails a softening of the cell and a reduction in the sensation of stiffness. In a groundbreaking investigation, it was shown that mesenchymal stem cells (MSCs) alter their morphology and develop towards lineage-specific differentiation when grown on matrices that exhibit varying, physiologically important matrix stiffnesses [39]. Cancer cells are capable of undergoing malignant transformation and taking on a mesenchymal phenotype.

Cav1 acts as a small oligomeric scaffold protein that is generally necessary for the development of membrane curvatures in structures like caveolae [40,41]. In addition, Cav1 connects to multiple other proteins, regulates the homeostasis of cholesterol, and controls a multitude of cell processes, including endocytosis, internalization of receptors, accumulation of cholesterol, cell signal transduction pathways, proliferation, and cell death [41,42]. Cav1-driven regulation of the signaling cues is critical in cancer. An example of this is the crosstalk of Cav1 and Rho GTPases, including RhoC, that promotes the emergence of metastases through inducing the expression of α5 integrin and the Src kinase-facilitated activation of the p130Cas/Rac1, FAK/Pyk2, and Ras/Erk1/2 signal transduction pathways [43,44]. It is important to note nonetheless that Cav1 also acts as a tumor suppressor by helping E-cadherin in the sequestration of β-catenin, thereby impairing the β-catenin/Tcf-Lef-dependent transcriptional activation of genes encompassing survivin, cyclooxygenase-2, cyclin D1, and multiple other proteins fostering the development of cancer [45,46]. Therefore, Cav1 fulfills functions of opposite outcomes, such as tumor suppressor and promotor in cancer, which have been reviewed in [41,47,48].

Cav1 suppresses tumor formation via regulation of contractile tension in epithelia [49]. In specific detail, caveolin’s contractile tension control is required to remove oncogene-transfected cells through the process of apical extrusion. Lack of caveolin-1 enhanced steady-state tensile stresses within epithelial monolayers, and consequently, the lack of Cav1 in the epithelial cells encircling oncogene-expressing cells hampered their apical extrusion.

Contractile force at the adherent junctions (AJs) mirrors the action of the actomyosin cortex, which is linked to E-cadherin-driven cell-cell adherence [50]. To comprehend how caveolae modulate junctional tension, first the actomyosin cortex at AJs was analyzed. Unexpectedly, the concentrations of non-muscle myosin II and phosphorylated myosin regulatory light chain (pMLC) concentrations at the junctions of Cav1-KD cells could not be determined even though mechanical tension was elevated. This suggests that although myosin II is required for tension, alterations in this motor cannot straightforwardly account for the elevation of tension within Cav1-KD cells. Neither the expression of E-cadherin nor the dynamics of the junctions were changed. In confocal microscopy, the F-actin values were slightly but persistently increased. This indicates that an actin modulatory signaling mechanism could be in charge of the junctional tension enhancement in Cav1 KD. 

To this end, the architecture of the F-actin network was primarily characterized using structured illumination microscopy (SIM). Apical junctional F-actin occurred with greater condensation in Cav1-KD cells, and fewer superimposed filaments and bundles were visible after the skeletonization of the microscopic images. Thus, the junctional cytoskeletal organization was modified through the knockdown of Cav1. This assumption was confirmed based on the measurement of the nematic order of F-actin at the junctions following the Fourier transformation of the fluorescence signals obtained from SIM images [51]. The nematic order coefficient was decreased in Cav1-KD cells, which is in line with a stronger co-linear or bundled arrangement of actin filaments. Ultimately, actin dynamics were characterized based on the expression of G-actin labeled with photoactivatable GFP (PAGFP-G-actin), and its fluorescence loss after photoactivation (FLAP) was assessed at the junctions. The T_1/2_ half-life of fluorescence decay was markedly elevated at the junctions of Cav1-KD and Cavin1-KD cells relative to controls, indicating that the F-actin pool remained more stable. Altogether, these results imply a mechanism that functions to stabilize F-actin and facilitate its bundling at AJs to be hyperactive in Cav1 KD cells. Formins seemed to be appealing contenders for imparting these effects, as they facilitate the assembly and stabilization of non-branched F-actin structures and operate at cell-cell junctions [52,53,54]. Similarly, the broad-spectrum formin inhibitor SMIFH2 applied to Cav1-KD cells restored F-actin levels at the junctions and normalized F-actin architecture and dynamics. It is important to emphasize that SMIFH2 compensated for the increased junctional tension observed in Cav1-KD cells. This suggests that an excessively active formin is likely in charge of the elevation of junctional tension when Cav1 is knocked down. Caveolae have lately been identified as mechanically active organelles that contribute to the mechanical damage protection of tissues, such as mechanical stress [55,56]. An understanding of this protective mechanism emerged from the finding that the caveolae become flattened when membrane tension rises. It was suggested that a membrane reservoir is thereby freed to buffer the changing membrane tension in a passive manner [57,58,59,60]. These results highlight another way in which caveolae can impact the mechanics of epithelial monolayers. In this case, the removal of caveolae enhances cortical contractility through stimulation of a phospholipid signal transduction pathway that aims at the actin cytoskeleton. It is proposed that caveolae modify active tissue tension by limiting the activity of this signaling mechanism. In this way, caveolae contribute to establishing an acceptable mode of epithelial tension for oncogenic cells that can be eradicated through apical extrusion. Cav1, known to be suppressed in a variety of cancer cells and oncogene-transformed cells, controls the mechanical phenotype [61,62]. Cav1-driven elevation of RhoA activity and Y397FAK phosphorylation guided actin cap production, which was positively associated with cell elasticity and stiffness perception in fibroblasts. Ha-RasV12-induced cell transformation and alterations in mechanical phenotype can be reverted by re-expression of Cav1 and mimicked when Cav1 is silenced in normal fibroblasts. Finally, this study revealed a novel function of Cav1 and identified a connection between mechanical phenotype and the transformation of cells. Consequently, mechanical properties can also be used as indicators of cell transformation. In summary, the role of Cav1 in cancer, in particular the comprehension of the canonical (Cav1 located in the plasma membrane) and non-canonical pathways (Cav1 situated in organelles and exosomes), is connected to the protein’s double function as tumor suppressor and facilitator of metastasis [63].

The general heterogeneity of tumors argues against the universal nature of the characteristics of cancer cells. Cell shape heterogeneity was found to be more tightly coupled to the mechanical state of the cells. However, single cells in multicellular spheroids display a lower level of mechanical heterogeneity than individual cells grown in monodisperse 3D culture systems. The reduced heterogeneity among cells found in spheroids implies that there is mechanical cooperation among the cells that comprise a solitary spheroid. Another possibility for the reduced heterogeneity within multicellular spheroids lies in their rather simple composition and architecture. There are also more intricate multicellular models available, such as organoids or even better tumoroids, which means tumor-like organoids. Organoids comprise three-dimensional complex ex vivo tissue cultures that can be obtained from embryonic stem cells, induced pluripotent stem cells, or tissue-resident progenitor cells. They have spatially limited lineage connectivity and higher-order self-organization, which renders them attractive quasi-physiological models [64]. Organoids still lack the full composition of cells, molecules, and factors that reside within a patient’s solid tumor. Although organoid cultures are superior to spheroid culture models, they still have weaknesses. Therefore, it is advantageous to generate samples directly from freshly isolated (resected or biopsy) patients’ cancer tissue and to preserve the complete structural organization of the TME and the ECM [65].

Organoids provide a biologically meaningful stage for enhancing translatability. Co-cultures represent no novel concept in the experimental work, as they are frequently utilized to investigate interferences between epithelial cells and other relevant cell populations, for example, lymphocytes, neurons, and blood vessels, as recently reviewed in [66]. The cultivation of epithelial cancer organoids with immune cells has provided valuable findings on the pathogenesis of various cancers, and the opportunity to genetically engineer these types of organoids in the absence or presence of immune cells offers a distinct and pertinent model for the examination of carcinogenesis [67,68,69]. The co-culture of mouse tumor organoids and adipocytes delivered new findings on colorectal cancer. For example, it has been demonstrated that adipocytes stimulate the proliferation and dedifferentiation, which is evidenced by elevated Lgr5 and CD44 and reduced mucin-2 and sucrase-isomaltase mRNA expression levels, of colon cancer organoids [70]. The researchers also hypothesize that adipocytes act as a metabolic regulator and energy supplier to support the growth of colon cancer cells, which is a candidate mechanism to account for the association between obesity and colon cancer. The ECM is not a mere passive spectator in cancer biology; nevertheless, the biological implications are frequently not investigated or considered in conventional laboratory experiments [71]. Co-culture experiments can solve this problem. Established organoids of pancreatic ductal adenocarcinoma, for instance, normally form ductal and basement membrane architectures, but this structure is destroyed upon co-culture with pancreatic stellate cells within a collagen matrix, resulting in deterioration of the basement membrane and enhanced invasion of the collagen matrix [72]. In addition, co-cultivation of pancreatic cancer organoids together with both stromal and immune cells results in the activation of myofibroblast-like cancer-associated fibroblasts, a phenomenon that was not evident in 2D cell culture models [73]. A model system that enables the interplay between cancer cells, stromal cells, and immune cells is consequently crucial for examining the pathogenesis of cancer.

New techniques are constantly being developed to optimize organoid cultures, such as the incorporation of self-generating hydrogels consisting of an ECM extracted from human tissue in place of mouse Matrigel. For instance, a methodology has been developed for preparing extracts from the ECM of breast mammary glands that can undergo spontaneous gelling and produce hydrogels [74]. The important point is that these hydrogels sustain biological signaling reactions that differ between cancer and normal epithelial organoid cultures [74]. Culture systems with air-liquid boundaries, in which the basal surface of the stem cells is in direct physical exposure to the culture medium and the apical surface is in contact with air, are equally interesting. This arrangement can more precisely reproduce the characteristics of the TME in specific types of cancer, like the luminal surface of colorectal carcinoma [75].

Based on the issues discussed, the question can be asked whether it is likely that cancer cells have the same mechanical properties from patient to patient, regardless of the type of cancer and the variables. However, the stage of cancer seems to have an influence on the mechanical phenotype, as there appears to be a strong correlation. The majority of research carried out to date has shown that individual cancer cells are softer compared to healthy cells, as illustrated in several review articles [76,77,78], which all focus on AFM stiffness analysis. The differences in the stiffness of cancer cells with varying invasive abilities, however, are subject to less agreement.

Recently, it has been revealed that mechanical characteristics at the level of the cell (stiffness, viscoelasticity) and at the level of the plasma membrane (fluidity) are interlinked [79]. More invasive cancer cells have been shown to either soften with magnetic tweezers with fibronectin-coated superparamagnetic beads, such as ovarian cancer cell lines [80] and with AFM, such as ovarian HEY, HEY A8, OVCAR-3, and OVCAR-4 cancer cells [81] and B16 melanoma cell variants [82], or stiffen, such as prostate, liver, and breast cancer cell lines [83,84,85,86], during the course of cancer advancement. Moreover, even when breast cancer cells are measured in an adherent and non-adherent state using AFM, there is still the finding that the softer breast cancer cells are more invasive into 3D collagen matrices and cause more fiber displacement when invading these 3D collagen fiber scaffolds [87]. Some of the observed inconsistencies may be attributable to the considerable diversity of cancer cells and the signaling pathways participating in the process of invasion [88]. Within this, higher stiffness (apparent elastic modulus and E_0_) was observed in cells with a mesenchymal phenotype and the highest migration performance (SW480), and lower stiffness in cells with an epithelial phenotype and low migration performance (HT29) [79]. Cell height and power-law exponent were increased in the softer HT29 cells, which is consistent with earlier findings [89,90]. The Newtonian viscosity coefficient η of the utilized viscoelastic model exhibited a positive correlation with stiffness. This result is not entirely clear, as it could be related to the viscosity of the cytoplasm in the vicinity of the cortex, which is also investigated at the penetration depths used (800–1500 nm, typically) [79]. However, there is currently no further data to substantiate this conclusion, and further research is needed to make a statement [79].

## 4. Malignant Cancer Progression (Cancer Metastasis)

Cancer that has disseminated to other areas of the body, which is commonly referred to as metastatic disease, is the leading contributor to the majority of cancer deaths. Cancer cells surmount numerous hurdles, such as surveillance by the immune system, to propagate to secondary sites effectively [91]. A plethora of research performed over the last two decades heavily implies that mechanical forces are also implicated in cancer progression and responses to traditional therapeutic regimens [92,93,94]. These forces also include fluid mechanics, which are increasingly coming into focus. On their route to establishing a metastasis, cancer cells and factors released by the cancer utilize and harness three key body fluids: the blood, lymph, and interstitial fluid [95,96,97]. The important fact is that fewer than 0.01% of the thousands of cancer cells penetrate the bloodstream and survive to develop metastases [98,99]. Thus, it is not likely that all cancer cells can be cleared from the patient by surgery. Thus, it is critical to comprehend the specific steps of the metastatic cascade to develop inhibitory treatments. All of these steps are concerned with mechanical encounters between the cancer cells and the various microenvironments they experience during metastasis [95]. Circulating tumor cells (CTCs) and their derived products, comprising soluble factors, cell-free DNA, and extracellular vesicles (EVs), can migrate directly through the hematogenous circulation [91,100] or sequentially exploit both the lymphatic and vascular systems to populate distant organs [101,102,103].

This idea that fluid mechanics can influence metastasis goes back to an early investigation that pioneered the “hemodynamic theory” and demonstrated that arterial blood flow in specific organs is positively correlated with the occurrence and patterns of metastasis [104], which demonstrates a connection between fluid mechanics and the secondary location of metastasis.

During transportation in liquids, CTCs are exposed to and act on different mechanical forces, which can affect their destiny in a variety of ways. For example, high shear forces acting on CTCs can trigger mechanical stress that results in cell fracture and death [105], while intermediate shear forces have been demonstrated to promote the intravascular dormancy phase and the extravasation of CTCs [106]. A better understanding of the mechanical forces to which CTCs and tumor-associated material are subjected in fluids is therefore critical to completely unraveling the metastatic cascade and defining susceptible CTC stages for therapeutic interference. This investigation reveals a novel mechanism by which a VEGF-VEGFR2-AKT-ATOH8 signaling axis induced through cyclic laminar shear stress (LSS) confers survival to mimic circulating tumor cells (m-CTCs) [107]. 

### 4.1. Increased Vascular Permeability around Cancers

The effect of enhanced permeability and retention (EPR) involves the extravasation of blood components from leaking tumor-induced vessels and their retention in the TME, thereby increasing interstitial pressure. Cancer cells can easily enter the leaky vessels to spread to targeted tissues and organs. As expected, these suspended cells with a CK8^+^/CD45^−^/DAPI^+^ phenotype have been seen in blood vessels and are referred to as m-CTCs. Quantitative polymerase chain reaction, western blotting, and immunofluorescence were employed to assess the alterations in gene expression of m-CTCs that exhibit sensitivity to LSS pacing. In addition, the expression of atonal bHLH transcription factor 8 (ATOH8) in CTCs of 156 CRC patients and mice was investigated using fluorescence in situ hybridization and flow cytometry [107]. The m-CTCs actively reacted to LSS by inducing the expression of ATOH8, which is a fluid mechanosensor involved in intravascular surveillance and the plasticity of metabolism. Notably, ATOH8 was observed to be upregulated through activation of the VEGFR2/AKT signal transduction pathway facilitated through LSS-triggered VEGF secretion. ATOH8 subsequently transcriptionally induced HK2-driven glycolysis, thereby enhancing the intravascular survival of colorectal cancer cells within the blood circulation.

The onset of fluid mechanics was the detection of diffusive transport of oxygen around blood capillaries, which was reviewed in [108]. Pioneering contributions have been identified that advance the application of fluid mechanics concepts to accompany genomic and molecular signaling investigations for cancer research [109,110,111,112,113]. The fluid mechanics of the cancer microenvironment were also explored in the work by Maeda and colleagues [114]. With the concept of the enhanced permeability and retention (EPR) effect, Maeda and colleagues [114] advocated the critical contribution of fluids extravasating from the tumor vasculature leading to elevated interstitial pressure, which constitutes a dominant element in the TME and a governing mechanism for cancer treatment.

### 4.2. Bodily Fluids

Blood, lymph, and interstitial fluid properties can be characterized based on their biophysical attributes, which are affected by their unique constitution and features, providing an appreciation of the mechanical stress that each fluid can exert on CTCs and other tumor-derived constituents [115,116]. These interconnected partitions exhibit various flow modes (which rely on the magnitude of the dimensionless Reynolds number, among other factors) and flow velocities that are encountered or utilized by the CTCs and/or tumor-secreted matter during transport. In the lymphatic system, for example, it is a generally laminar, pulsating flow with low amplitude, which is primarily powered by viscosity and exhibits low velocities [117]. Conversely, blood possesses a far higher density of circulating particles (blood cells and other factors) and has faster flow rates because of the heart’s pumping capacity. Moreover, the blood flow in arteries can be pulsating with high amplitude and turbulent flow, while the flow in veins is generally laminar [118]. In addition, the biophysical cues vary based on the flow and type of vessel, as well as the nature of the organ. In total, CTCs and tumor-derived material moving in the circulatory system are subjected to shear rates ranging from around 10 s^−1^ in the lymph [117] to around 1000 s^−1^ in large arteries [118,119].

Blood and lymph play a part in the interstitial fluid flow that typifies cancer [94]. The interstitial fluid produced due to high capillary blood pressure and cellular pressure within the solid primary tumor [120] is dissipated through the lymphatic system and its primary valves, which are under lower pressure. This directional shift was characterized in a model [121] on the principle of Darcy’s law and is primarily related to the variations in pressure, surface area, and hydraulic conductivity among these networks. The interstitial fluid pressure (IFP), intriguingly, not only eases the spreading of cancer cells and tumor-associated material [96], but also has key implications for the infiltration of drugs into the primary cancer site [94].

### 4.3. Tumor Interstitial Fluid Facilitates the Cancer Cell’s Migration and Invasion

Solid primary cancers create a complex microenvironment comprising cancer cells, stromal cells, the ECM, blood vessels, and lymphatic vessels. When cancers grow to 1–2 mm, they need to become vascularized to elevate oxygen levels, and hence they need to recruit nearby blood vessels or induce the growth of new blood vessels [122]. The growth of the primary solid tumor depends on angiogenesis, which offers oxygen and required nutrients through the de novo formation of new blood vessels, and lymphangiogenesis, which functions in the elimination of excessive fluids and the dehydration of cancer cells and tumor-secreted factors released from the tumor [123]. Moreover, both fluid systems enable the transportation of immune cells into and out of the tumor, and the lymphatic circulatory system directs local immune surveillance through the oversight of adjacent lymph nodes [124].

The Darcy model accounts for variations in IFP inside the tumor that promote the diffusion and convection of fluids from the blood into the outflowing lymphatic system [96]. In primary tumors, the IFP gradient is exacerbated by the incompleteness of the immature tumor-associated blood vessel system, which contains many “holes” [125]. Moreover, fast tumor growth leads to massive solid stress, which subjects the tumor-associated vascular network to additional vasoconstriction due to compression and tension [126,127]. Thereby, a high IFP is produced inside the tumor tissue. In an autochthonous mouse model for pancreatic ductal adenocarcinoma, the intratumoral IFP was over nine times higher compared to the IFP found in equivalent healthy tissue [128,129]. Even though a high IFP value is evident in the core of the tumor [128], the IFP value decreases at the tumor periphery, resulting in an interstitial fluid flow across the peritumoral stroma into the lymphatic vessels [96]. The interstitial fluid flow consists of both convection and diffusion fluxes in the direction of the periphery [130], with velocities of about 0.001–0.004 mm s^−1^ [131]. At the same time, the stress puts pressure on the blood and lymph vessels [132], which impedes the oxygen supply and homeostasis of the solid tumor. Conversely, cancer cells are prone to enhance the liberation of pro-angiogenic factors [120], resulting in an abnormal and hyperpermeable blood circulation (low velocity and very heterogeneous), which further promotes the malignant response of cancer cells.

A typical case of vascular flow and transport is presented in, in which scientists explain a conceptual approach for incorporating a discrete 1D model of tissue vasculature into a 3D continuum model of interstitial trafficking [133]. Fluid shear flow across a meshwork of vessels is characterized by Poiseuille’s law, which links blood flow to channel radius, pressure, and the viscosity of the blood liquid. The transfer through vessel walls can be characterized by Starling’s law, which links the rate of extravasation to vessel permeability and the pressure mismatch between the vessel and the tissue. The speed and direction of the interstitial flux can be determined by Darcy’s law, which connects these variables with the pressure gradient and the tissue’s hydraulic conductivity. To maintain the unique architectural structure of the vascular system, an approach associated with a continuous 3D model of the interstitial space that is not limited to a spatially averaged variable was established. These three essential relations are illustrated in Figure 3, which are employed to determine the characterization of intravascular and interstitial flow. Poiseuille’s and Starling’s laws are integrated into this drawing (Figure 3). Poiseuille’s law (Equation (1)) inks the intravascular flux $Q_{v}$ (Figure 3, gray arrow) to the radium of the vessel *R*, the dynamic blood fluid viscosity *µ*, and the intravascular pressure gradient $p_{v}$. Starling’s law (Equation (2)) links the rate of extravasation $J_{v}$ (Figure 3, yellow arrows) to the hydraulic conductivity of the vascular wall $L_{p}$, the vascular surface area *S*, the reflection coefficient *σ*, the vascular oncotic pressure $\pi_{v}$, and the interstitial oncotic pressure $\pi_{i}$. Darcy’s law (Equation (3)) connects the interstitial flow speed $u_{t}$ (Figure 3, orange arrows) to the interstitial tissue hydraulic conductivity k and the interstitial pressure gradient $p_{t}$. These three linkages can be found within the overall published literature on the physical modeling of tumor-based vascular flux and angiogenesis.

Poiseuille’s law is provided in the following Equation (1):(1)$$Q_{v}=\frac{\pi R^{4}}{8\mu }\;\nabla p_{v}$$

Starling’s law is described in Equation (2) below:(2)$$J_{v}=L_{p}S\left(\left(p_{v}-p_{i}\right)-\sigma \left(\pi_{v}-\pi_{i}\right)\right)$$

Darcy’s law is given in Equation (3) as follows:(3)$$u_{t}=-k\nabla p_{t}$$

The interstitial fluid of cancer cells promotes the migration and invasion of cancer cells. The interstitial convection flux holds the capability to accelerate the invasion of glioblastoma cells and to promote the migration of amoeboid human breast carcinoma cells towards the lymphatic drainage system, where cancer cells can exit the primary tumor with the support of macrophages [134]. Interstitial fluid flow alters stromal cells by enhancing the polarization of macrophages, which encourages the migration of cancer cells [135]. In a mouse model of breast cancer, migrating macrophages are transformed into sessile perivascular macrophages [136]. However, the effect of the interstitial flux is still elusive. As collagen I stimulates mesenchymal motility, cancer cells may migrate in the opposite direction of convective flux. The flux of interstitial fluid interacts with the flux of luminal vascular tissue to promote the intravasation of cancer cells [137]. The motion of the interstitial fluid affects the migration of cancer cells toward the lymphatic and blood vessels. It has been hypothesized that convective pressures associated with lymphatic and blood circulation push tumor-related contents and cells to aid their targeted spread (as well as the spread of soluble melanoma factors, or EVs), toward the vascular events or ECM at the circumference of tumors. There is not enough scientific substantiation for this hypothesis. Other mechanisms, like postnatal angiogenesis (the formation of new blood vessels through recruited endothelial progenitor cells from the bone marrow) and vasculogenic mimicry, are possibly accountable for the formation of blood vessels [138].

## 5. Chromosomal Instability, Exosomes, and Cell-Free DNA Foster Cancer Metastasis

Cancer metastasis is the hallmark of cancer, accounting for the majority of cancer-related deaths. However, it is still not clearly revealed. Invasion marks the first point in the metastatic cascade, when cancer cells gain the capacity to migrate, invade nearby tissue, and penetrate lymphatic and blood vessels to spread. A contributing part of genetic alterations in invasion is not generally agreed upon. The skeptical argument is that cell motility depends exclusively on external stimuli like hypoxia, chemoattractants, and mechanical factors such as the stiffness and/or viscoelasticity of the ECM. In tumor hypoxia, the content of oxygen is lowered from 4.6% to 9.5% in healthy tissues to less than 1–2% [139]. However, there is growing evidence that mutations can initiate and enhance the migration and invasion of various types of cancer cells. The published literature on the implications of chromosomal instability and genetic mutations on the migration and invasion of cancer cells is presented in a recent review [140]. Chromosomal instability, release of exosomes, and cell-free DNA (cfDNA) of primary tumors, normal cells, or cancer cells have emerged as hallmarks for the malignant progression of cancers. All of these features appear to be triggered by mechanical influences such as interstitial flow, external forces, and/or stiffness. As cancer cells are subject to mechanical cues, chromosomal instability may enable cancer cells to adjust to different environmental conditions by potentially adjusting their mechanophenotype and may help them to disengage from the primary tumor and subsequently metastasize. The stiffness-driven release of exosomes can reduce the response of immune cells by provoking their apoptosis. The cfDNA, which can also be released from exosomes, can protect cancer cells in the vascular system from destruction by the immune response, as the specific receptors existing on the cancer cells cannot be identified. Finally, these events can all contribute to cancer metastasis.

### 5.1. Chromosomal Instability

Cancer cells start to spread from the primary tumor before the first steps of the invasion-metastasis cascade take place [141]. The metastatic cascade is the result of chromosomal instability due to ongoing errors in the separation of chromosomes at the time of mitosis (Figure 4). Errors in chromosome segregation lead to disruption of micronuclei and release of genomic DNA into the cytosol, which consequently activates cytosolic DNA-sensing signaling pathways (cyclic GMP-AMP synthase stimulator of interferon (IFN) genes) and subsequent nuclear factor-κ-light-chain-enhancer of activated B (NF-κB) signaling downstream pathways [142]. Research indicates that the type of primary seeding cancer cell dictates the varying metastatic characteristics in terms of growth and responsiveness to treatment [143,144]. In vivo and in vitro trials demonstrate that metastatic cancer cells migrate alone [145]. In humans, the assumption is that cancer cell seeding involves the collective action of a cluster of cancer cells migrating in concert [146], which is the timepoint at which epithelial-mesenchymal transition (EMT, see Section 5) comes into play.

Nuclear abnormalities comprise small nuclei (micronuclei) that harbor intact chromosomes or fragments of them. These micronuclei can cause intricate chromosome rearrangements during cancer development and progression [147]. Nuclear mechanophenotype alteration, such as nuclear envelope blebbing (softening), can induce invaginations that engulf actin and intermediate filaments [148], which result in epigenetic modifications. In addition, chromosomal abnormalities can lead to polyploidy in cancer cells and, subsequently, to different daughter cancer cells with different mechanophenotypes, which metastasize to varying degrees.

### 5.2. Exosomes

Exosomes comprise a class of small extracellular vesicles that are liberated by all kinds of cells. They exhibit a size spectrum of 30–200 nm [149]. Exosomes are formed when the boundary membranes of the endosomes expand towards the lumen and generate multivesicular endosomes. When multivesicular endosomes are trafficked toward the cell surface and merge with the plasma membrane, their intraluminal vesicles are secreted by the cells in the form of exosomes [150,151,152]. Exosomes act in intercellular communication between cancer cells and their microenvironment via the exchange of information through the cargo of the recipient cell, comprising proteins, lipids, DNAs (ssDNA, dsDNA, mtDNA), RNAs (mRNA), and microRNAs (miRNA, long non-coding RNA) [153]. Exosomes can prevent their cargo, such as miRNAs, from being destroyed by environmental cues, and thus exosome miRNAs can be maintained within human blood plasma and other bodily fluids. Thus, these exosome miRNAs may serve as a non-invasive biomarker for cancer prognosis, treatment, and monitoring, including cervical cancer and malignant glioma [154,155]. Exosomes, which are extracellular vesicles holding genetic material, proteins, and lipids, also play a pivotal role in the creation of the pre-metastatic niche [156]. Increasing stiffening of the TME triggers the release of exosomes from cancer cells via the Akt pathway and Notch pathway activation [157]. Stiffening of the ECM is associated with activation of Akt, in turn stimulating GTP charging of the small GTPase Rab8, which ultimately propels the release of exosomes [157]. Exosomes derived from cancer cells exposed to a stiff ECM stimulate tumor growth in an efficient manner. Proteomic profiling revealed that the Notch signaling pathway is induced in cells subjected to exosomes of cancer cells cultured on stiff ECM [157]. Luminal soluble proteins and membrane proteins of exosomes, such as lipid-anchored proteins, molecules bound to the periphery of the membrane, and transmembrane proteins, can be released into the TME [158]. The membrane proteins of the exosomes permit them to target specific cells. When they are attached to the targeted cells, exosomes can initiate signaling via receptor-ligand engagement and become endocytosed and/or phagocytosed. Exosomes are able to merge with the membrane of the target cell, thereby liberating their cargo into the cytosol and altering the physiological condition within the cell [159]. Thereby, the exosomes of a particular cancer cell can impact the performance of surrounding cells, the cellular environment, and the phenotype of remote cells and tissues, which has a strong capacity for systemic implications [160]. For example, exosomes of cancer cells can impede the immune response against themselves, primarily by causing apoptosis of T lymphocytes [161]. Exosomes of pancreatic cancer cells target T lymphocytes and interfere with their gene expression profile, e.g., genes such as ATF4, MAPK, and EIF2α, which are involved in the onset of apoptosis, and consequently compromise their antitumor capabilities [162].

Exosomes can also be captured by adjacent or distant cells, where they are involved in the post-transcriptional regulation of gene expression through the targeting of mRNA. Exosomal miRNAs may fulfill various purposes, such as involvement in inflammatory responses, cell migration, proliferation, apoptosis, autophagy, and epithelial-mesenchymal transition [163,164]. Once the exosome miR-132 is captured by endothelial cells, the expression of RabGAP-P120 is decreased through signaling, which encourages the tubularization of endothelial cells [165]. 

Exosomes originating from the primary tumor possess a set of integrins on their outer surface that control adhesion of exosomes to specific cell types; this may permit cancer cells to bind the ECM or translocate its cargo into recipient cells to establish organtropism [166]. New evidence indicates that tumor-derived exosomes can trigger the creation of pre-metastatic niches that facilitate the progression to metastatic disease [167,168]. Importantly, exosomal tumor RNAs are found to activate the Toll-like receptor 3 (TLR3) of the alveolar epithelium to activate chemokines (CXCL1, CXCL2, CXCL5, and CXCL12) that are key for neutrophil recruitment and pre-metastatic niche creation inside the lung [169]. Tumor-derived exosomes may also induce N2 polarization in neutrophils to drive the migration of gastric cancer cells [170] and neutrophil extracellular traps (NETs) and cause cancer-related thrombosis [171]. Moreover, exosomes can transport PD-L1 from the primary tumor to other locations in the organism to inhibit the immune defense in the pre-metastatic niche [172]. In an appropriate pre-metastatic space, cancer cells need to undergo an angiogenic switch to attract various cells that alter the surrounding tissue and create an environment that eases the growth and spread of metastases [173]. While organ tropism is poorly comprehended, the study of exosomes and pre-metastatic niche creation is advancing knowledge in this field. Targeting exosomes or any other factors that are critical for pre-metastatic niche establishment may be able to avoid the seeding of metastatic cancer, but there is still a long way to go before the formation of pre-metastatic niches is completely clear.

### 5.3. Cell-Free DNA (cfDNA)

Cell-free (cf) DNA can be released by exosomes that emerge increasingly after a stiffness increase in tumors. As an alternative option to liquid biopsy, circulating cfDNA is steadily secreted from clonal cancer cells into the bloodstream [174,175]. Variations in cfDNA can be detected by ultra-deep NGS even at extremely low frequencies (<1%) and have been employed for the early identification of relapses in different tumors (e.g., colorectal cancer, pancreatic cancer, neuroblastoma) and recently also in premalignant lung and bladder forms of the disorder [176,177,178,179]. Currently, several platform technologies offer adequate sensitivity in detecting circulating tumor (ct) DNA to accurately diagnose lung tumor patients who recur within one year of subclone detection and for accurate screening of premalignant cervical cancer [176,180]. However, methods for identifying and profiling ctDNA need to be enhanced, and methods should have improved sensitivity and more specificity with quantitative thresholds to prevent overdiagnosis. An important biological constraint is the quantity of ctDNA collected from early-stage cancer patients, such as when there is even less than 0.1% of ctDNA identified in plasma using digital droplet PCR or NGS techniques [181]. The enrichment process has to be conducted on the basis of biological or physical characteristics. The results are distorted by cancer-related mutations, which are not limited to cancer patients, and the presence of clonal changes in blood cells resulting from aging and clonal hematopoiesis, which are both considered critical issues [182,183]. Genomic driver alterations of cancers in the BRAF, CDKN2A, EGFR, FGFR3, HER2, NF1/2, PIK3CA, RAS, and TP53 genes may also be detected in non-tumor probes. In addition, identifying the tissue of neoplastic lesion origin can be very difficult [178]. Therefore, the restrictions on the clinical benefit of ctDNA analysis are obvious. Apart from cancer diagnosis, cfDNA can protect cancer cells from being identified and, hence, destroyed by the immune system.

## 6. Altered Mechanical Cues in Primary Tumor Tissues Establish the Tumor Microenvironment (TME)

Solid tumors and their accompanying TME are composed of cancer cells and stromal elements, comprising the ECM, basement membrane (BM), vasculature, immune cells, and fibroblasts (Figure 5). As the tumor advances, all elements alter their physical appearance and functionality [184,185,186]. In many types of cancer, although there are a handful of exceptions, primary tumors are usually mechanically stiffer than their healthy source tissue [185,187,188,189]. For instance, human breast tumors are five times stiffer than healthy tissue, and this high level of stiffness is positively related to malignancy [190]. Mammary tissue from mice with tumors is 24 times stiffer compared to healthy mammary tissue [191]. The stiffness of human liver tissue has a positive relationship with the incidence of hepatocellular carcinoma, with a threshold value of 20 kPa [192]. In addition to general stiffening, heterogeneity of intratumoral stiffness represents another salient mechanical feature of tumor tissue [193]. The measurement with ultrasound elastography reveals the enormous spatial variability of tissue stiffness in breast and liver tumors [194]. In human breast cancer biopsies, the periphery of the primary tumor is 7-fold stiffer (E = 5.51  ±  1.70 kPa) compared to its center (E = 0.74  ±  0.26 kPa), whereas healthy breast tissue exhibits a stiffness of 1.13–1.83 kPa [195]. In addition to stiffness, the viscoelasticity of cancerous tissue also distinguishes it from that of normal tissue. For instance, in vivo assessment using magnetic resonance elastography (MRE) indicates that the fluidity of human benign meningioma tissue remains 3.6 times higher than that of aggressive glioblastoma tissue. This solid-like characteristic of glioblastomas eases their aggressive infiltration into the adjacent tissue [196].

The increased stiffness of the cancer tissue is primarily attributable to over-deposition and enhanced cross-linking of the ECM, particularly of collagen [197]. The TME is constantly reorganized by tumor and stromal cells and delivers physicochemical information to control the gene expression and functioning of these cells through the activation of a range of intra- and extracellular molecular receptors and signal transduction pathways, including integrin, PIEZO ½, and Rho/ROCK. These receptors pick up extracellular biophysical cues and transmit them to the cell nucleus. They then transmit intracellular feedback to the restructuring of the extracellular TME [198,199]. The constitution, stiffness, and organization of the ECM define its regulatory function in the progression of tumors. The ECM consists of fibrous proteins, glycoproteins, polysaccharides, and proteoglycans [200]. High expression of different ECM proteins is associated with a worse outcome in various types of cancer [197]. Aberrant expression of ECM enzymes, like matrix metalloproteinases (MMPs), which control ECM restructuring, is an indicator of a negative prognosis [189,198]. As the principal structural elements of the ECM, collagens account for up to 60% of the mass of the tumor and the stiffness of the tumor tissue [199,200]. High collagen levels encourage the formation of breast cancer and invasive phenotypes [201]. The ECM is a three-dimensional network that, besides the main constituent collagen, is composed of other macromolecules, including elastin, fibronectin, glycoproteins, hyaluronic acid, laminin, lysyl oxidase (LOX), proteoglycans, and tenascins, which are upregulated in cancer and thus provide structural, mechanical, and biochemical cues to cells such as cancer cells and CSCs [202]. All this modifies the mechanical properties, such as increasing stiffness, of the in vitro collagen matrixes [203] and TME of solid tumors [204]. The stiffness of the ECM has a decisive impact on the transformation, proliferation, and motility of cancer cells. For instance, high ECM stiffness eases the positioning of the yes-associated protein (YAP) in the cell nucleus, which is necessary for the transformation of normal, healthy breast cells triggered via the RTK-Ras oncogene [205]. Human breast cancer cells have a higher concentration of miR-18a in the stiffer ECM, which favors the growth of cancer cells [206]. Stiff ECM enhances breast cancer cell growth and invasion by producing high cell tension. High stiffness of the ECM upregulates TWIST1, thereby enhancing EMT and consequently promoting metastasis of these breast cancer cells [207]. High ECM stiffness in pancreatic ductal cancer cells triggers the signal transducer and activator of the transcription 3 (STAT3) signaling pathway, which enhances matricellular fibrosis and ductal epithelial tension and drives progression of the tumor via decreased transforming growth factor-β (TGF-β) signaling and elevated activation of β1-integrins [208]. High ECM stiffness and cell contractility enhance the MMP activity of pancreatic cancer cells three to tenfold, which facilitates migration, invasion, and angiogenesis [209]. During the progression of various solid tumors, sequestration, restructuring, and networking of the ECM constitution alter and cause the stroma to stiffen in a gradient formation going from the tumor periphery (highest stiffness) to the tumor core (lowest stiffness). Since the center of the tumor mass is less rigid, the cancer cells migrate durotactically outward to higher stiffness, which leads to metastatic cancer cells. The spatial distribution of liver CSCs corresponds to the stiffness of the tumor tissue, such as when the circumference of the tumor is 13 times stiffer and comprises 13 times more CSCs compared to the center of the tumor [210]. Thus, there are indications that the plasticity of cancers can result in a dynamic alteration of the frequency of CSCs. Matrix stiffness also has an impact on CSC function, as it causes the activation of mechanosensitive cell surface receptors that subsequently activate mechanosensory and/or mechanoregulatory molecules including integrins, vinculin, talin, paxillin, FAK, and YAP [211]. The mechanoregulatory molecules can alter both cancer cells and CSCs by impacting various signal transduction cascades [212]. For example, FAK is able to trigger via the integrin-FAK-Src signal transduction pathway the activation of serine/threonine-protein kinase (AKT), β-catenin (Wnt signaling pathway), cyclin D1, ERK (Ras-ERK signaling pathway), JNK (RhoA-JNK pathway), phosphatidylinositol-3-kinase (PI3K), and other proteins, whereas tumor suppressing genes like phosphatase and tensin homolog (PTEN) and glycogen synthase kinase 3α/β (GSK3α/β) are blocked [213,214]. In addition, CSCs can transmit mechanical signals via the RhoA/Rho-associated protein kinase (ROCK) signaling pathway [35]. Beyond that, tumor stiffness impacts cancer cells, CSCs, and stromal cells through activation and nuclear translocation of the transcriptional activators YAP1 and WW domain-containing transcription regulator 1 (WWDR1) (TAZ) (Hippo pathway) [205,215]. 

In reaction to increased ECM stiffness, glioma cells trigger Piezo1 activation in focal adhesion sites and enhance calcium influx; thereby, the integrin-FAK signal transduction is activated and ECM stiffening is even further enhanced [216]. High tissue stiffness stimulates the Rho/ROCK signaling pathway to enhance actomyosin-based cell tension and collagen sequestration, which reinforces tissue stiffness [217]. In addition, the stiffness of the tumor tissue affects the morphology of the vessels, the vascular barrier performance, and the vascular integrity [218,219,220,221]. For instance, FAK activity triggered by the stiffness of the matrix stimulates Src and high levels of phosphorylated vascular endothelial cadherin (VE-cadherin) at endothelial cells’ adherent junctions [220]. Enhanced stiffness of the ECM causes improved angiogenic dispersion and leakage, which unwantedly encourages the dissemination of cancer cells into the vascular system [218]. Even though the density and the level of cross-linking of the collagens determine the stiffness of the ECM, they can impact angiogenesis in opposite directions. In an in vitro 3D organ culture system model of sprouting angiogenesis, enhanced matrix density decreases angiogenesis and vascular meshwork development, probably due to the fact that a stiff ECM is more difficult for endothelial cells to reshape [222]. Enhanced collagen cross-linking encourages the angiogenic sprouting of the spheroid and improves the stiffness of the substrate [189,218]. Especially in the area surrounding the tumor vessels, the ECM is thicker and increasingly linearized [223]. These results indicate that the impact of enhanced ECM density and alignment, including cross-linking and linearization, on the stiffening of the primary tumor and tumor progression appears to be mutually exclusive. The impact of ECM stiffness on angiogenesis relies on cell-ECM adhesion. On 2D polyacrylamide (PA) gels, which are coated with type I collagen, a softer ECM (200 Pa for soft ECMs vs. 10 kPa for stiff ECMs) stimulates endothelial cell loop generation, mimicking the onset of angiogenesis, but shifts to a suppressive mode when collagen concentration is decreased from 100 μg/mL to 1 μg/mL.

## 7. Impact of Tumor Microenvironment (TME) Stiffness on Cancer Advancement

The impact of ECM stiffness on tumor progression is, nevertheless, debatable. Ovarian cancer cells, for instance, tend to be more invasive when they are in a softer environment [224]. In contrast, higher matrix stiffness in solid tumors is linked to enhanced invasion and metastasis, which is attributed at least in part to the rise in CSC population and biomarkers [225,226]. There is evidence that matrix stiffness is able to activate receptors and mechanosensor/mechanoregulator proteins like integrin, FAK, and YAP, which modulate the features of cancer cells and CSCs via various molecular signal transduction pathways. A sudden shift to a low ECM stiffness has been found to sustain stem cell formation of malignant tumor repopulating cells (TRCs) or CSCs in a soft ECM of 90 Pa but not in a stiff ECM of 1.05 kPa [212,227,228,229]. At the edge of the softer core of a primary tumor, cancer cells undergo cell death and necrosis, which is referred to as necroptosis. Consequently, the core region of primary tumors becomes softer. Thus, at the boundary between a softer tumor core and a stiffer tumor belt, the cancer cells of the core undergo an increase in stiffness that can induce changes in these cancer cells towards an invasive phenotype, enabling metastatic progression of the cancer. In addition, epigenetic modifications triggered by extracellular or intracellular factors can prepare cells for effective escape from the primary site and encourage the development of disseminated cancer cells that can settle at distant sites and create metastatic entities. 

It has been revealed that the compliance of a magnetic platform, which displays high ligand tether mobility, increases the stemness and tumorigenicity of cancer cells [230]. CD133^+^ liver CSCs soften regional niches to sustain their stemness, increase resistance to medication, and alleviate metastasis [231]. These varying reactions to ECM stiffness could be attributed to the reliance of mechanosensing on the particular cancer type, the heterogeneity of TME, and the heterogeneity of cancer cell subpopulations. Collagen, as a fibrous substance, exhibits the properties of strain hardening, nonlinear elasticity, and natural anisotropy [232,233]. Stiffening of the ECM can be induced by collagenous stretch-hardening in the presence of even low levels of cell contraction-induced stretch and, conversely, promotes cancer propagation when stretch-hardening is not reversible [234,235]. The nonlinearity of collagen fibers, such as compressive buckling and tensile stiffening, has been observed in a finite element model to ease the transmission of mechanical cues over long distances of about nine cell lengths to faraway cells [236]. In addition to stiffness, ECM architecture, like fiber alignment, interconnectivity, porosity, and topography, is also responsible for the invasive phenotypes of cancer cells, comprising locomotion, protrusions, and MMP activity in self-assembled 3D collagen matrices [237,238]. Elevated collagen density can decrease ECM pore size, and an intermediate pore size of 5 to 12 μm is regarded as an effective enhancer of glioma invasion [239]. By utilizing an interconnecting matrix of hydrogels from 30 Pa to 310 Pa, the impacts of pore size and stiffness on cancer cells have been separated [240]. The boundary in the pores increases the polarization, traction, and migratory velocity of cancer cells. The speed of cell migration correlates positively with the stiffness of the ECM within the constrained ECM, whereas this relationship is biphasic in the unconstrained ECM [5,241]. In breast cancer, collagen fibers that orient themselves vertically to the border of the primary tumor are observed to enhance invasion and promote metastasis [242]. A high degree of collagen cross-linking, along with enhanced ECM stiffness, eases cancer cell invasion via increasing integrin-regulated FAK-Src signal transduction [211].

Apart from the matrix mechanics and acellular elements of the stroma surrounding the tumor, the stroma provides a harbor for various types of immunoregulatory cells, like fibroblasts and endothelial cells, that supply oxygen and substances to sustain and support the development, advancement, invasion, and metastasis of the cancer via stromagenesis and angiogenesis [243]. In colorectal cancer (CRC), the metastatic capacity depends heavily on the stroma. Fibroblasts in the stroma, the most prominent cell type of the stromal cell community [244], are key contributors to stromal cross-talk and cancer advancement. A subset of these fibroblasts undergoes activation to form cancer-associated fibroblasts (CAFs), which are synonymously referred to as myofibroblasts and express alpha-smooth muscle actin (α-SMA) [244,245,246,247,248]. Fibroblasts become CAFs when activated through TGF-β [248,249]. These CAFs promote a compacted myofibroblastic moiety and the accumulation of ECM proteins accompanying cancer fibrosis [250]. CAFs produce an accelerating force, resulting in their enhanced cell stiffness on 2D soft substrates in vitro [251,252]. In vivo, CAFs secrete growth factors that enhance metastatic advancement of cancer cells [253], and they synthesize collagen and the collagen cross-linker (LOX), both of which stiffen the ECM and reshape the ECM’s constitution and organization [204]. The contractile force of the CAFs can also alter the biomechanical characteristics of the tumor by causing compressive stresses in the tumor and gradients of pressure on the proliferating epithelial cancer cells due to mechanical nonlinear stress-strain deformation [132,254]. Extra stress is created through the growth of cancer cells in a restricted spatial environment [254]. Hence, stromal cells and CAFs are crucial biophysical participants in cancer propagation. The mechanics of stroma are especially pertinent to intestinal and pancreatic cancer.

## 8. Intravascular Spread of Cancer Cells

The evolution of secondary tumors in a section of the body far away from the original primary cancer is referred to as metastasis. Although metastasis is the main contributor to cancer treatment breakdown and death, it is still barely explained. Cancer patients release a high number of cancer cells into the bloodstream every day; melanoma research in animal models, nonetheless, indicates that less than 0.1% of cancer cells metastasize [255]. The evolution of metastases demands that the cancer cells abandon their primary site, travel in the bloodstream, withstand the pressure in the blood vessels, adapt to a new cellular setting in a secondary location, and avoid the deadly battle with immune cells [91,256]. According to Hanahan and Weinberg, “activating invasion and metastasis” constitute a hallmark of cancer [88]. In fact, invasion of adjacent tissues and dissemination to remote sites to generate metastases is a key symptom of the malignancy of cancer (Figure 6). Finally, metastasis is the leading reason for death in over 90% of cancer patients [257]. Improving knowledge of the dynamics of this process will contribute to identifying targets for molecular treatments that can stop or possibly even reverse cancer growth and progression. 

When cancer cells infiltrate the vasculature through intravasation, they are typically eliminated through shear stress or immune surveillance. Fewer than 0.01% of cells that manage to escape a primary tumor undergo extravasation when they approach a secondary target site [258,259,260]. Intravasation can be of two different types, such as active and passive (Figure 6) [261]. During the process of passive intravasation, the majority of cells either die or undergo apoptosis [261]. It is assumed that these cells are shed due to the waning nutrient delivery caused by the hypoxic environment of the tumor and the leaky blood vessel circulation [261,262]. In active intravasation, the cells travel along nutrient and growth factor gradients towards a blood vessel by the mechanism of chemotaxis [263,264]. These cells are able to break down the ECM and the basement membrane and actively penetrate into a blood vessel [261]. Alternatively, these cancer cells can squeeze through the ECM and the basement membrane pores [265,266]. In the bloodstream, these cancer cells combine with blood platelets, enabling the cancer cells to resist the shear force [258,267]. The process of inducing EMT (see Section 11) in these circulating cancer cells enables the rearrangement of intermediate filaments to resist this sheer force [268]. Cancer cells can evade elimination through the immune system by various mechanisms (Figure 7), such as coating with immune cells, such as macrophages, platelets, or neutrophils, or masking their receptors by the secretion of broadly immunosuppressive cytokines. 

The liberation of soluble factors like VEGF, IL-10, TGF-β, prostaglandin E, and Fas out of cancer cells supports the establishment of an immunosuppressive environment [269,270,271,272,273]. These factors are secreted by tumor-associated cells through the increased stiffness of primary tumors and their TME. For instance, VEGF release causes the enrolment of immature dendritic cells and macrophages [274,275]. Tumor-associated dendritic cells and tumor-associated macrophages (TAMs) inhibit the capacity of mature dendritic cells and macrophages to eradicate cancer cells via inhibiting T cell activation and phagocytosis [269,276]. The expression of inhibitory receptors, including programmed death receptor-1 (PD1) and its ligands PD-L1 and PD-L2, by cancer cells, such as CTCs, impairs the activation of T lymphocytes [277]. In addition, antigen presentation, especially by the major histocompatibility complex (MHC), is diminished on the cancer cell surface, enabling them to circumvent immune surveillance [278]. In addition, platelet coating can protect circulating cancer cells against natural killer cells (NK cells) and T cells, and platelets can transfer MHC to cancer cells and thus mislead the immune system [279]. Adherent platelets can encourage the shedding of the NKG2D ligands MICA and MICB from the surface of cancer cells via ADAM10/17-facilitated cleavage (Figure 8) [280,281]. Moreover, platelet-coated cancer cells exhibit less detectable CD112 and CD155 on their cell surface, which act as ligands for the activating NK cell receptor DNAM-1 [281], which further reduces the immune response toward cancer cells. The coating of cancer cells with platelets is referred to as cloaking. Immune checkpoint proteins depress the immune response as they inactivate immune cells competent to destroy tumors. Therefore, treatments with immune checkpoint blockers can bypass immune escape by boosting T cell-based clearance of cancer cells [282,283]. Consequently, the blockade of immune checkpoints has revealed impressive outcomes in a broad range of solid cancers, like melanoma [284,285] and non-small cell lung cancer [286].

## 9. Dual Functions of Tumor-Associated Macrophages (TAMs)

Macrophages are double-edged swords among myelomonocytic cells with dual capacity in cancer, which mirrors their plasticity as a reaction to environmental circumstances [8,287,288,289]. The switch between the two main states of macrophages is induced by increased stiffness of the TME. Macrophages are able to eradicate cancer cells, promote antibody-dependent cellular cytotoxicity and phagocytosis, induce vascular injury and tumor necrosis [290], and trigger mechanisms of tumor resistance facilitated through innate or adaptive lymphoid cells. In sharp distinction, macrophages in the majority of established tumors promote cancer progression and metastasis through various mechanisms, among them fostering the survival and proliferation of cancer cells, angiogenesis, and the suppression of innate and adaptive immune mechanisms [291,292,293,294,295]. These diverse functions of macrophages can be explained by their mechanical stimulation by the primary tumor and/or the TME. Thus, a change in the mechanophenotype of TMEs can induce a switch in TAMs.

TAMs have emerged as a prototype for the interplay between inflammation and cancer [296]. Macrophages play an essential role in the anti-tumor effects of chemotherapy, monoclonal antibodies (mAbs), and radiotherapy [297,298] by facilitating anti-tumor activity and triggering adaptive immune reactions. In addition, they are an attractive target of modern checkpoint blocker immunotherapies, as they express inhibitory counter-receptors (such as PDL1 and PDL2) and therefore inhibit adaptive immune reactions [290,299,300,301,302]. As TAMs exhibit pro-tumoral functions and immunosuppressive activity, they may aid or hinder cancer cell intravasation and survival (Figure 9). 

In a similar fashion to TAMs, neutrophils can act as a double-edged sword [303]. In human peripheral blood, neutrophils are the largest leukocyte subgroup and play an essential part in the immune reaction to infectious pathogens. It is also hypothesized that the opposite functions of neutrophils can be due to their altered mechanophenotype induced by the elevated TME stiffness. Due to their short longevity and terminal differentiation, they have for a long time been assumed to perform a marginal function in cancer-related inflammation. Recently, however, animal models have demonstrated that tumor-associated neutrophils (TAN) may also polarize into various phenotypes in reaction to tumor-derived excitatory stimuli, including mechanical cues. TAN can therefore exercise pro- and antitumor responses (Figure 10) [304,305,306].

### 9.1. TAMs Foster the Intravasation of Cancer Cells

Another crucial step in metastasis is that cancer cells squeeze through small gaps in the vascular endothelium to obtain entry into the host vasculature [307]. An experiment with intravital multiphoton imaging provided direct and kinetic imaging of the intravasation. Based on this experiment, an intravasating cancer cell is consistently chaperoned by a macrophage within a single cell diameter, providing direct proof of the involvement of TAMs in the intravasation of cancer cells [308,309]. Clinical observations have concordantly revealed the tripartite organization of TAMs, cancer cells, and endothelial cells as the TME of metastasis. This specific metastatic environment is a marker for enhanced hematogenous metastasis and a bad outcome, at a minimum, in breast cancer [310]. The mechanisms behind this synergistic encounter are intricate. On the one hand, macrophages degrade the ECM surrounding the endothelium through a series of proteolytic enzymes like cathepsins, MMPs, and serine proteases [311,312,313]. Activated TAMs have a direct impact on the enhancement of metastasis by directly secreting soluble factors [314]. M2 macrophages can disrupt the matrix membrane of endothelial cells through the deposition of MMPs, serine proteases, and cathepsins and break down different collagens and other constituents of the ECM, thereby promoting the migration of cancer cells and tumor stromal cells [315,316]. In addition, cytokines secreted by cancer cells that have undergone EMT (see Section 11) also stimulate the differentiation pathway of TAMs, creating a positive feedback circuit between TAMs and EMT [317]. On the other hand, TAMs channel cancer cells into the bloodstream through a positive feedback circuit comprising CSF-1 secreted by the cancer cells and EGF secreted by TAMs [318]. The first-named cytokine promotes macrophage movement and stimulates EGF secretion, which in turn sends messages to the cancer cells and facilitates chemotactic migration in the direction of the blood vessels [318,319]. Thus, hampering either the CSF-1 or the EGF pathway interferes with the movement of both cell types and also decreases the circulating tumor cell count.

### 9.2. TAMs Promote Cancer Cell Survival Inside the Circulation

As soon as the cancer cells have invaded the vasculature, they must be prepared for survival and exit from the bloodstream. Clots packaged around cancer cells decrease survival stress by NK cells in response to tissue factor (TF) in the bloodstream and in the capillaries [320,321]. A strategy in which macrophage functions were perturbed using genetic techniques reduced the survival of cancer cells in the pulmonary capillaries and prevented tumor invasion into the lungs, notwithstanding the development of blood clots, suggesting an instrumental involvement of macrophages in this process [322]. Two conceivable mechanisms could be responsible for this phenomenon. A recent investigation revealed that the recruited macrophages induced the PI3K/Akt survival signaling pathway in newly spread breast cancer cells by activating vascular cell adhesion molecule-1 (VCAM-1) through α4 integrins [323,324]. Activation of the PI3K/Akt survival pathway thereafter protected the cancer cells against proapoptotic cytokines, such as TNF-related apoptosis-inducing ligand (TRAIL) [323]. In addition, numerous cancer cells protected by the chemokines or cytokines released directly from the macrophages manage to survive [322].

### 9.3. Role of Macrophages in the Presence of Flows

Several studies have demonstrated that tumor interstitial fluid fosters the migration and/or invasion of cancer cells. For example, interstitial convection flow can enhance the invasion of glioblastoma cells [325] and induce the amoeboid migration of breast cancer cells [134,326] in the direction of the lymphatic drainage, in which cancer cells can evade the primary tumor with the assistance of macrophages [136]. The interstitial fluid stream can also impact stromal cells, for instance, by encouraging macrophage differentiation into polarized populations, which, for their part, can enhance the directional migration of cancer cells [327]. As innate immune cells, macrophages are capable of vastly adopting a wide range of phenotypes that are evoked by cues macrophages sense in their local microenvironment. In one extreme, these functionally plastic cells can become polarized into an inflammation-supporting M1 phenotype, whereas in the other extreme, an immunosuppressive M2 phenotype arises [328]. Macrophages in TME frequently exhibit M2 status, and the amount of M2-like macrophages invading close to or into tumor tissue has been determined to have a correlation with poor prognosis [329]. In fact, M2-like macrophages within tumors can secrete various growth factors and cytokines that encourage metastasis by facilitating the invasion and intravasation of cancer cells and fleeing the immune system [328,330]. Conversely, M1-like macrophages in tumors are hypothesized to induce tumor suppression through activation of anti-tumor immunity [331]. The opposing roles of the two subtypes of macrophages could be influenced by the TME and represent a potential target for tumor therapy [332].

Migratory macrophages switch to sessile perivascular macrophages and assist cancer cells during intravasation in a mouse model of breast cancer [333]. However, whether this mechanism can be impacted by interstitial flow is uncertain. Convective flow is a stream that is directed down a gradient. For example, a convective flow can be a pressure gradient in the case of a primary solid tumor. Cancer cells can migrate in opposition to convective flow by utilizing supports like collagen I fibers, which can facilitate the acquisition of mesenchymal phenotypic locomotion [334]. An extensive in vitro investigation also demonstrated that interstitial fluid flow can interact with luminal vascular flow to increase the intravasation of cancer cells into lymphatic vessels [335]. However, it is unclear whether this also takes place in vivo. The interstitial fluid flow is able to channel cancer cells and tumor-associated factors into the vicinity of lymphatic or blood vessels by manipulating the route of cancer cell trafficking. In fact, it is reasonable to speculate that such convective forces, in conjunction with blood and lymphatic circulations, dislodge cancer material and cells. Thereby, the metastatic spreading of cancer cells is fostered, as is the propagation of soluble cancer factors, or EVs, in the direction of the vasculature or ECM at the circumference. At the same time, therapeutic delivery is somehow restricted within solid tumors [120].

Macrophages support cancer cells in the primary tumor by facilitating cancer cell growth, migration, and invasion. They are referred to as TAMs. TAMs have been found to release EGF and travel in streams together with cancer cells in the direction of blood vessels [319,336,337]. In the perivascular area, a subpopulation of macrophages expressing Tie2^high^, together with stationary cancer cells in a network termed the TMEM gateway, performs an essential function in cancer cell intravasation. These macrophages facilitate intravasation through the formation of a regional, temporary aperture in the blood vessels [338]. Perivascular macrophages also trigger cancer stem cells and the dormant phase that primes cancer cells to settle and persist in remote organs [339,340]. Additional evidence pointing to the significance of macrophages in metastases highlights the necessity of colony-stimulating factor 1 (CSF-1) for metastatic propagation, an integral component of macrophage survival and proliferation [341,342]. In addition, a special macrophage population, the so-called metastasis-associated macrophages (MAMs), which are labeled by the surface markers F4/80^+^/CSF-1R^+^/CD11b^+^/Gr1-CX3CR1^high^/CCR2^high^/VEGFR1^high^, are attracted to the lung and are relevant for the amount and extent of metastases in the experimental PyMT metastasis model [343,344]. Moreover, MAMs are necessary for effective metastatic sprouting and have a distinct expression profile that favors their aggregation and metastatic colonization [345,346].

Intravital time-lapse recordings have already demonstrated that the interference between cancer cells and macrophages increases considerably when cancer cells extravasate [347]. This encounter took place even though 70% of the cancer cells were retained in the vessels. When the cancer cells were trapped while extravasating (traversing the vessel), the macrophages also interacted directly with the extravasated portion of the cancer cell [348,349]. In addition, macrophages connected to cancer cells stretched out incredibly long, whereby they formed fine pseudorods that could not be detected at a lower resolution [348].

## 10. Extravasation and the Function of Microchannels

Cancer cells have to accomplish the challenging process of extravasation, which comprises attachment to endothelial cells at the secondary target region, modification of the endothelial barrier (or interface), and transendothelial migration into the subjacent tissue, before they can metastasize [276]. Similar to TAMs and TANs, the endothelium can act as a double-edged sword. The endothelial cells can act either as a promotor, induce cancer cell transmigration and invasion into the ECM, or act as a barrier [5,350,351,352]. The predominant type of extravasation is paracellular migration, where cancer cells pass through between two adjacent endothelial cells [276]. Throughout this process, numerous ligands and receptors, among them selectins, cadherins, and integrins, promote adhesion between the cancer cell and the endothelial cells, which is extensively time-regulated [353,354]. Moreover, extravasation is based on the cooperation between cancer cells and blood cells, comprising platelets, myeloid-derived suppressor cells (MDSCs), and TAMs [276]. Platelets act to initiate an invasive mesenchymal phenotype through the liberation of TGFβ1 and ATP from granules, thereby inducing endothelial junction modification and enhancing transendothelial migration of cancer cells [276,355,356]. Myeloid cells upregulate VCAM1 and VAP1 on TAMs, which then secrete VEGF to enhance vessel permeability [345]. Cancer cells also develop invadopodia on their basal surface as they acquire mesenchymal characteristics. These structures are protrusive and adhesive and secrete matrix metalloproteinases like MMP-9 and MMP-2 to break through the endothelial barrier [357,358]. After successful spread, disseminated cancer cells acquire a cell plasticity that promotes their survival by escaping the immune system and subsequently promotes the overgrowth of metastases.

### Microchannel Structures Caused by Macrophages

The presence of membranous connections between cells in vivo was first proven in 2008, when membranous nanotubes between immune cells and stromal cells were seen within the mouse cornea [359]. There are similar structures between perivascular macrophages and residing tissue cells, as well as between pericytes in discrete capillary networks [360,361]. Membrane-like interconnections have been reported in several types of cancer [362]. In glioblastoma, these structures contribute to chemoresistance, and analogous structures have been found in MDA-MB-231 brain metastases with the aid of light sheet microscopy [363,364]. These structures are frequently designated by various names, including membrane nanotubes [359], tumor microtubes [363], or tunneling nanotubes (TNTs) [362,365]. These structures differ according to their diameter, their cytoskeletal constituents, and whether they are closed or open. All of them, nevertheless, are classified under the wide category of thin membranous connections (TMC). These structures have been demonstrated to be able to facilitate various processes such as the replacement of mitochondrial and nuclear elements, the enhancement of growth factor responses, and the exchange of ions [364,366,367,368]. All these various processes are facilitated through the formation of direct cell-to-cell connections and remote cell-to-cell interactions [369]. M-Sec (TNFAIP2) is a key regulatory factor of TNT-like membranous linkages found in macrophages, cancer cells, and other cell types [370,371]. M-Sec belongs to a 73 kDa cytosolic protein with an N-terminal polybasic domain that is involved in the recruitment of M-Sec to the plasma membrane and a C-terminal domain that engages RalA, thereby controlling the exocyst function necessary for TMC generation [372,373]. Earlier work has demonstrated that TNTs originating from macrophages increase cancer cell invasion in vitro and in a zebrafish-based invasion model [374].

Although macrophages are implicated in the extravasation of cancer cells, the processes underlying extravasation are not yet fully understood. In an in vitro assay that imitates extravasation, TMCs from macrophages were found to be important in stimulating extravasation of cancer cells. The importance of macrophage TMCs could be verified **in vivo** with the aid of an M-Sec-deficient mouse in which the macrophages are faulty in TMC generation. In addition, high-resolution intravital imaging (IVI) revealed macrophage engagement with cancer cells in the extravasation phase and how a macrophage in the lung parenchyma interfaces with a cancer cell through a TMC before extravasation. The microenvironment may play a crucial role in this process. Unfortunately, most mechanistic in vitro investigations on TMCs have been carried out using 2D cell culture systems, whereby TMCs are developed on top of the substrate in a 3D space. The presence of cancer cell TMCs has been identified in 3D collagen fiber matrices [368] and in 3D bioprinted scaffolds [375]. These findings lead to the conclusion that the TME may serve as an important regulator of cancer cell processes that depend on TMCs, such as extravasation.

## 11. Epithelial–Mesenchymal Transition (EMT): What Is New?

The EMT has been known for quite a while, and it seems to be able to understand the process of cancer metastasis. Many regulatory factors have been identified as potential anti-cancer therapy targets. However, the states of EMT are not so clearly defined, and there seem to be a lot of intermediary states in this reversible process. Even new mechanical triggers for the EMT have been identified.

### 11.1. New View on Epithelial-Mesenchymal Transition

EMT refers to the transdifferentiation process by which transformed epithelial cells acquire the capacity to invade, withstand stress, and propagate [88,376]. Epithelial cells remain motionless and are closely attached to one another and to the adjacent ECM [377]. EMT controls the reversible biochemical changes that allow a given epithelial cell to adopt a mesenchymal phenotype and imparts epithelial cells with epithelial-mesenchymal plasticity [378], which is crucial for cancer advancement and metastasis (Figure 11). Not all cells derived from the primary cancer, of course, promote the formation of metastases. An investigation of the determinants of metastatic risk in a mouse model of breast cancer found that asparagine synthetase, which is a metabolic enzyme, is linked to the progression of metastases [379]. Reducing asparagine concentrations by ʟ-asparaginase treatment or food restriction reduced the spread of metastases. Thus, the availability of asparagine supported the EMT [379].

More recently, it is now widely recognized that the EMT program is a broad range of transitional stages in between the epithelial and mesenchymal phenotypes, as opposed to a history of progression that involves a dichotomous decision between a fully epithelial and a fully mesenchymal phenotype [380]. The transformation from one phase to the next is regulated through a series of growth factors [381] and signal transduction pathways [382]. In primary cancer cells, a spontaneous EMT alternates between various intermediate states with varying invasive, metastatic, and differentiation features [383]. Cancer cells with a mixture of epithelial and mesenchymal phenotypes are much more effective in circulation, colonization of the secondary location, and progression to metastasis [383]. In addition, transcriptional, chromatin, and single-cell RNA sequencing reveal that the various different states exhibit distinct cellular features, chromatin maps, and gene expression signatures that are governed by common and differing transcription factors and signal transduction pathways. Additionally, the distinct EMT phases are located in different microenvironments and are in close proximity to various stromal cells. [383]. For instance, metastatic cancer cells displaying the most distinctive mesenchymal phenotype multiply in the vicinity of endothelial and inflammatory cells. These cancer cells secrete large quantities of chemokines and proteins to recruit immune cells and encourage angiogenesis, thereby stimulating the formation of a unique inflammatory and highly vascularized cavity [383]. CAFs have also been found to stimulate and regulate cancer cell migration mediated by fibronectin alignment [384]. Moreover, hypoxia [385], metabolic stress agents, and matrix stiffness [386] induce the EMT shift within cancer cells. The transition is frequently controlled through transcription factors that are specifically programmed to repress epithelial genes and stimulate mesenchymal genes [387]. Epigenetic and post-translational regulators, like DNA or RNA methylation, as well as miRNAs, also have an important role to play in regulating the EMT mechanism [380].

Over the last few years, there has been an intense discussion about whether EMT plays a pivotal role in cancer metastasis and cancer chemotherapy resistance [382,388,389,390]. Lung and pancreatic cancer studies indicate that while EMT is not essential for metastasis, it still plays a role in chemoresistance [388,389]. The EMT not only impacts the biochemical phenotype of cancer cells; it is also impacting the mechano-phenotype of cancer cells. In specific detail, post-EMT interphase cells exhibit a softer actin cytoskeleton [391], which is in agreement with a study that revealed softening of adherent cells following EMT [392]. Nonetheless, more conclusive evidence is required to fully clarify the involvement of EMT in cancer progression and the metastatic cascade.

### 11.2. Non-Classical E-Selectin-Induced EMT

Although EMT may be necessary for the initial formation of metastases, the opposing mesenchymal-epithelial transition (MET) pathway is necessary for the progression of metastasis (Figure 11). In bone metastasis, E-selectin in the bone vasculature triggers the induction of MET and WNT in cancer cells to encourage the development of metastases [393]. E-selectin attachment activity, which is conveyed by the α1-3-fucosyltransferases Fut3/Fut6 and Glg1, is crucial in the development of bone metastases. In contrast to conventional EMT models, E-selectin-induced MET had no effect on the RNA expression of the main transcriptional regulators of EMT, including Snail1/2, Twist1/2, and Zeb1/2. In addition, EMT maker N-cadherin staining revealed no reduced expression but rather altered localization and altered apparent molecular weight, whereas the protein concentration of transcription factor Slug was significantly lower following E-selectin engagement. These observations, collectively, point to a non-canonical MET scheme that appears not to be the binary inverse of conventional EMT schemes. Extraction of the enriched nuclear genes in the “Sarrio EMT” [394] and “Hallmark EMT” gene panels revealed that the majority of the upregulated EMT-associated genes were implicated in immune-related pathways, whereas the downregulated genes involved were mainly released or extracellular proteins. Collectively, these findings suggest that engagement of E-selectin triggers a non-canonical MET-like switch in cancer cells. Ultimately, staining of epithelial markers demonstrated ubiquitous E-cadherin and intermittent EpCAM staining within BM2 bone lesions, which confirms the presence of MET throughout bone metastasis in vivo.

Numerous studies have demonstrated that EMT is frequently required to evade a primary tumor [395], and others have found that these cells must return to an epithelial state to repopulate an organ effectively [396,397,398]. The paucity of proof of how MET is derived, specifically in the context-dependent fashion necessary for metastatic colonization of a remote organ, has led to substantial debate in the field. Evidence indicates, however, that a unique underlying stromal cue—the binding to E-selectin via Fut3/6 and Glg1 that are expressed by bone metastatic cells—triggers MET to promote bone metastasis. The data also implies that this non-canonical MET operates Wnt signaling to stimulate stem cell generation through Sox2/9 expression and enhances the expression of Glg1 and Fut3/6.

E-selectin-triggered MET represents a non-canonical type of behavior in comparison to MET inducers like miR-200, which act on the main transcription factors of EMT [399]. Instead, a key role of E-selectin-induced MET involves activation of the Wnt signaling pathway, which is associated with self-renewal, CSC properties, and EMT initiation [400]. This proposed association between Wnt-induced stem cell formation and EMT is inconsistent with the necessity of MET in metastatic settlement, which thus poses an intriguing contradiction: How can MET and cancer stem cell formation coexist throughout metastatic dissemination? Coupling E-selectin-induced MET with Wnt signaling and Sox2/9 inducement revealed how E-selectin commitment could unravel this paradox during bone metastasis, which is similar to recent findings demonstrating that Prrx1 decouples EMT from stemness characteristics in lung metastasis [397]. The non-canonical EMT transcription factor paired-related homeobox 1 (PRRX1) overexpression activates EMT in specific cancers, comprising those of the stomach [401], colorectum [402], pancreas [403], or breast [397], and fosters a migratory and invasive phenotype. At a later phase of the metastatic progression, however, its expression needs to be silenced to encourage MET, metastatic colonization, and an epithelial phenotype exhibiting stem cell characteristics [397]. More specifically, two isoforms of PRRX1, PRRX1a, and PRRX1b play distinct roles in EMT and MET within pancreatic ductal adenocarcinoma [404]. PRRX1b enhances dedifferentiation, invasiveness, and EMT, while PRRX1a participates in differentiation and MET [404]. Alternative splicing generates two isoforms of the transcription factor paired-related homeobox 1 (PRRX1): PRRX1a and PRRX1b, which may be regulated by mechanical pacing during cancer cell metastasis. Moreover, alternative splicing may also be induced by hypoxia [405].

## 12. Conclusions and Future Directions

Throughout the complex process of the metastatic cascade, bidirectional interferences can be seen. These interferences seem to be key in regulating the malignant progression of cancers. The TME plays an important role in mediating the interplay between cellular and acellular components of the TME and cancer cells. These two new paradigms can strongly influence the clinical trials of new effective therapeutics against metastases. In addition, the mechanical cues of the TME, embedded stromal and immune cells, and the mechanical flow induced pressures on primary tumors and cancer cells. Moreover, epigenetic cues play a crucial role and can additionally impact cancer metastasis. The focus of the current vascular route for cancer cell metastasis may not be sufficient, as targeting embryonic factors linked to migration and identified during cancer metastasis may not be of special relevance for the alternative avascular route of pericyte migration/extravascular migratory metastasis. Thus, it may be possible that cancer cells can switch the traveling route for cancer metastasis to escape treatment.

Macrophages have emerged as a key regulatory player in the cancer cell intravascular metastatic cascade. However, TAMs fulfill contradictory functions in cancer metastasis. Similarly, TANs exhibit this contradictory behavior. Due to the heterogeneous nature of CAFs, they fulfill contradictory functions. For example, the multiplicity of CAF origins shapes the complexity of CAF biomarkers, and CAF subpopulations expressing diverse biomarkers are likely to exert divergent effects on tumor progression [406]. The interplay between cancer cells and other cell types, such as TAMs/CAMs and TAFs/CAFs, and the transition of neighboring normal cells into cancer-associated cells need to be explored more precisely from a physical viewpoint. These cross-talks can impact the outcome of cancer treatment. Apart from the cellular bidirectionality of cancer-associated cells, alternatively spliced transcription factors can act in a bidirectional manner. As the microenvironment, such as hypoxia or mechanical cues, can impact alternative splicing, it is necessary to re-analyze biochemical cues in a TME. The complexity of the TME needs to be rebuilt in experimental approaches by including embedded stromal and immune cells and endothelial vessels. Thereby, organoids with vascularization may be of benefit. The dynamics of TME and cancer cells need to be included in terms of phenotypic alterations and mechanical characteristic changes. Finally, the efficiency of cancer therapy and drug development may depend on the biochemical and mechanical complexity of the TME.

## Figures and Tables

**Figure 1 biomolecules-14-00184-f001:**
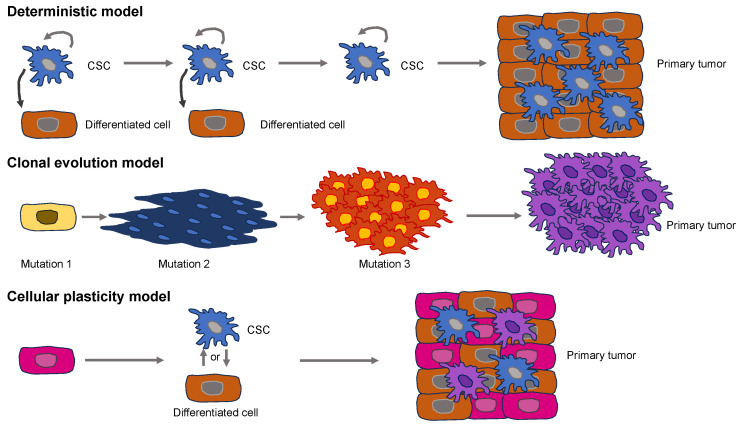
Models for the initiation of primary solid tumors. The deterministic model (**upper images**) presents CSCs at the apex of the hierarchical organization. CSCs possess the potential for self-renewal and can produce differentiated cells that are less cancerogenic. The clonal evolution model (**middle images**) accumulates genomic mutations or epigenetic alterations that enable cancer cells to proliferate faster compared to normal, healthy cells. The cellular plasticity model (**lower images**) postulates the idea that CSCs do not represent cells of origin. Instead, plasticity occurs in CSCs, or differentiated cells, that finally culminate in the primary tumor.

**Figure 2 biomolecules-14-00184-f002:**
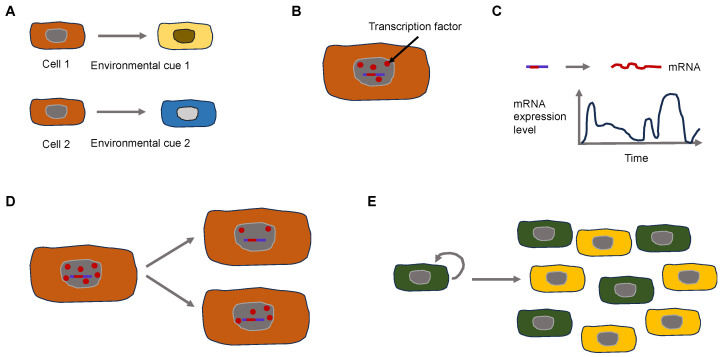
Extrinsic and intrinsic signals control the heterogeneity of cancer cells. (**A**) Extrinsic cues can lead to cell heterogeneity. Two isogenic cells are subject to different environmental cues due to broad microenvironmental variations that evoke various responses. (**B**–**E**) Intercellular variations lead to heterogeneity due to intracellular variations. (**B**) A small number of transcription factors (red dots) require different times to bind to the promotor region (short red line) on the DNA (blue line). (**C**) The activity of cellular processes such as transcription is time-dependent. (**D**) Non-equal cell division can cause intercellular differences. (**E**) Feedback circuits in cellular populations result in all (green) or none states (yellow) in cellular physiology.

**Figure 3 biomolecules-14-00184-f003:**
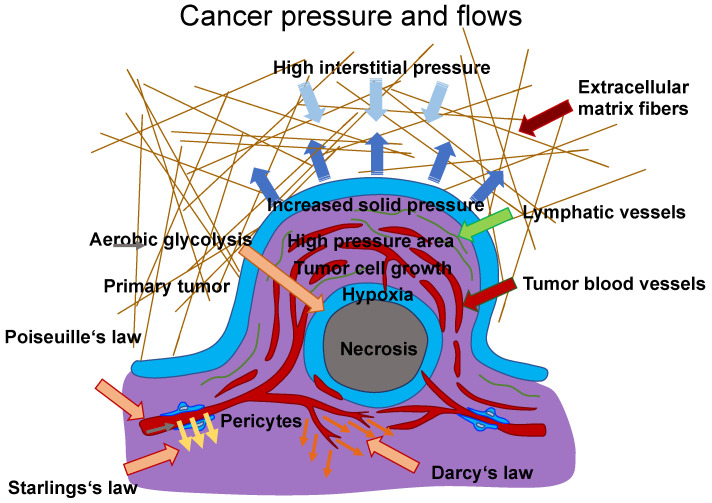
Model of flows between blood vessels and tissues. The Poiseuille (grey arrow), Starling (yellow arrows), and Darcy (orange arrows) laws can describe the flux within blood vessels and tissues. The extravasation flow can be characterized by Starling’s law, which takes into account the pressure on both faces of the capillary, such as vascular (arterial or venous) and interstitial pressure. The overall hydrodynamics can be described using hydraulic conductivity coefficients for the arterial and venous flows to determine the singular performance of the vascular resistance. Darcy’s law can be used as a model for the movement of cancer cells with both inhomogeneous and isotropic conductivity. Therein, the cancer cells are treated as a fluid with constant density. This “cancer cell fluid” flows through a porous environment, such as the ECM, which is defined as rigid and immobile. Thus, the porosity is constant. Simulations have shown that the tumor mass increases from regions with high conductivity to regions with low conductivity if the tumor form is not altered. Thus, Darcy’s law states that a higher flow velocity is present in areas exhibiting higher conductivity. Poiseuille’s law states that the flow is highest at the vessel wall due to the enhanced fluid shear stress, which has an impact on cancer cell survival within vessels. In addition, there is high interstitial pressure (light blue arrows) from the TME, which compresses the tumor mass. The dark blue arrows indicate the expansion pressure caused by the proliferating tumor mass in the direction of surrounding tissue. Hypoxic conditions prevail in the core of a tumor mass due to fewer blood vessels, and some tumor masses are necrotic in their core area. Aerobic glycolysis, which requires oxygen, is one of the hallmarks of cancer and can only take place where sufficient oxygen is available.

**Figure 4 biomolecules-14-00184-f004:**
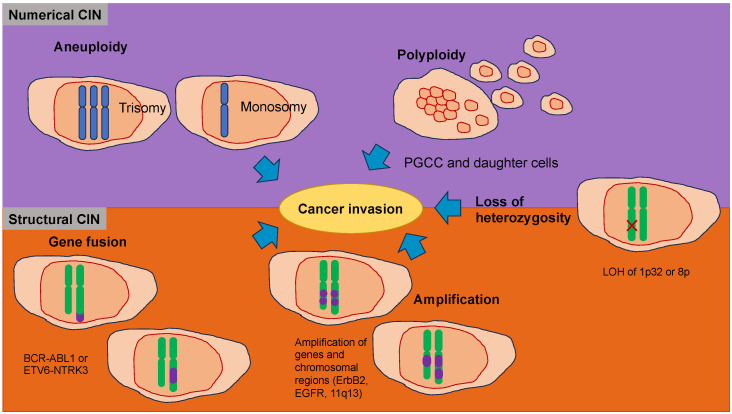
Chromosomal instability (CIN) represents a hallmark of cancers that function in the migration and invasion of cancer cells. CIN can be caused by a gain or loss of entire chromosomes, which is referred to as numerical CIN (lilac), or structural reorganization, which is referred to as structural CIN (orange). The loss of heterozygosity (LOH) contributes to numerical and structural CIN, whereby genomic alterations, such as allele loss, impact cancer cell invasiveness. Polyploidy is the existence of additional sets of chromosomes that consequently alter the genetic profile of cancer cells and increase their invasive capacity. Polyploid giant cancer cells (POCC) are seen in numerous cancers and indicate extreme tumorigenic, invasive, and metastatic capacity. Aneuploidy, such as loss of chromosomes (monosomy) or acquired chromosomes (trisomy), can impact cancer cell motility differently. Various gene fusions due to chromosomal rearrangements impact cancer cell invasiveness via different regulatory pathways and mechanisms. Amplification, which is a copy number enhancement of a specific region of the genome, causes elevated gene expression. In this case, when the specific gene is linked to cellular migration, it can increase cancer cell invasion.

**Figure 5 biomolecules-14-00184-f005:**
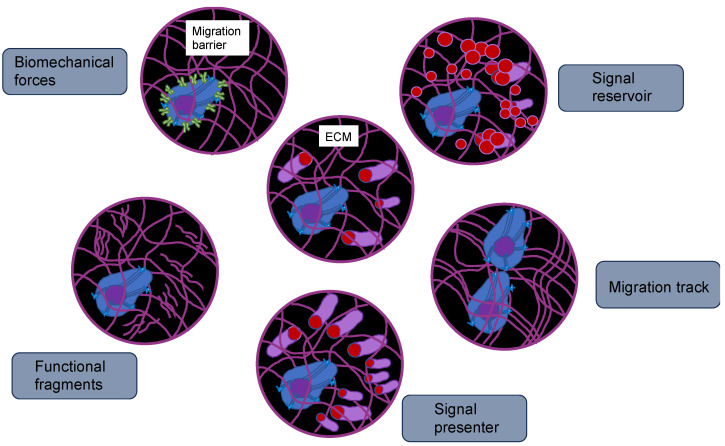
Functions of ECMs, such as TME. The ECM represents an attachment scaffold for cells that is needed for maintaining the tissue polarity and asymmetric division of stem cells. Due to its composition, it can foster or impair cancer cell migration. The ECM can sequester growth factors and hinder their free diffusion. There are also other constituents of ECMs that can tether to growth factors and act as co-receptors or signal presenters, which aid in determining the cell-cell interaction direction. The breakdown of the ECM by matrix metalloproteinases (MMPs) can modify the functionality of cancer cells. The physical characteristics of the ECM can be perceived through focal adhesions of cancer cells, which may result in various alterations of the cell phenotype, including the rearrangement of the 3D genome.

**Figure 6 biomolecules-14-00184-f006:**
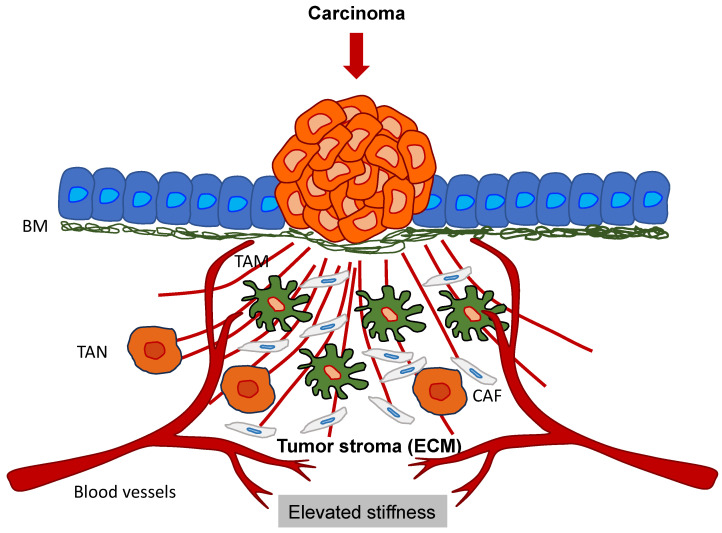
Primary solid tumors interact with the tumor stroma, such as the ECM scaffold, immune cells (TAMs and TANs), and stroma cells (CAFs). The cancer cells remodel the surrounding ECM, and consequently, the ECM stiffens.

**Figure 7 biomolecules-14-00184-f007:**
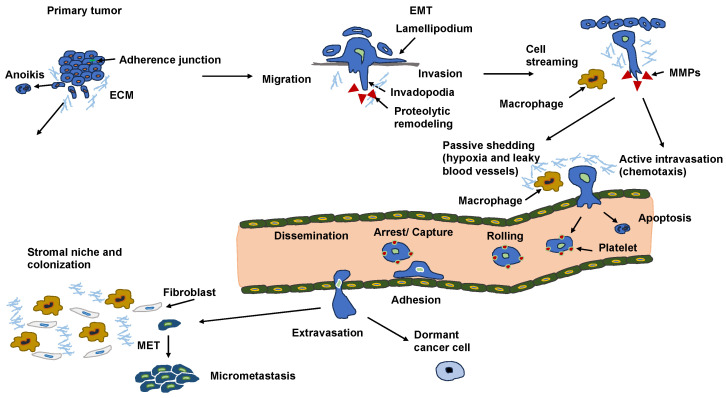
Consecutive steps of the metastatic cascade.

**Figure 8 biomolecules-14-00184-f008:**
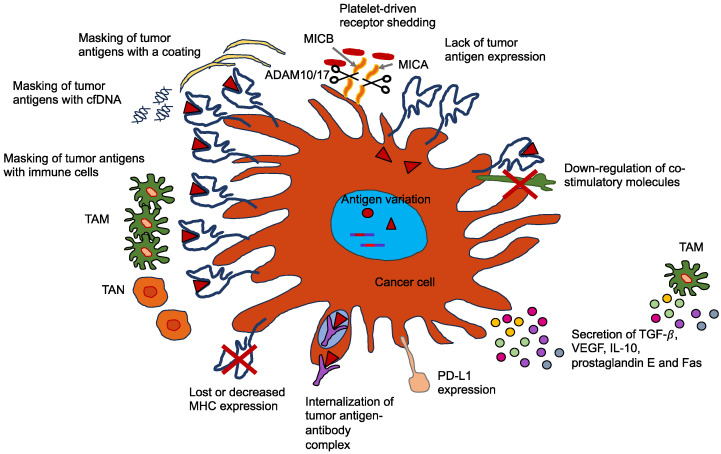
Cancer cells escape the immune response via various mechanisms.

**Figure 9 biomolecules-14-00184-f009:**
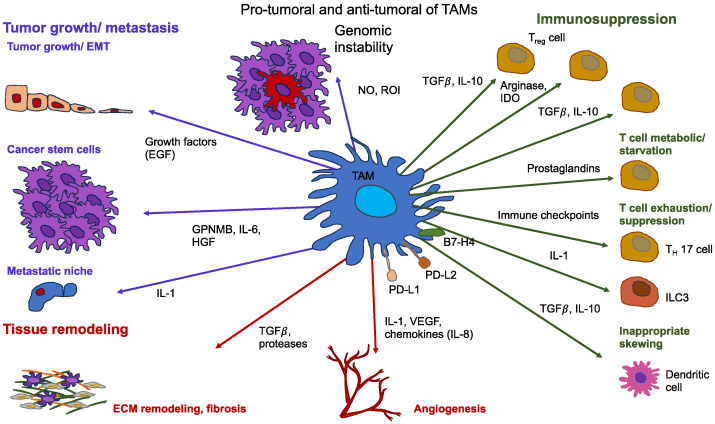
TAMs fulfill tumor-promoting tasks that help at various stages of tumor development. TAMs secrete nitric oxide (NO) and reactive oxygen intermediates (ROI), leading to DNA injury and genetic instability during the initiator stage of the primary tumor. TAMs secrete epidermal growth factor (EGF) and various mediators like hepatocyte growth factor (HGF), IL-6, and glycoprotein NMB (GPNMB), which promote the growth of CSCs. In later phases, TAMs participate in metastatic dissemination by liberating IL-1 and TGF-β, which are also implicated, along with various proteases, in reconstructing the ECM and pathological fibrosis. TAMs are an important supplier of angiogenic determinants: vascular endothelial growth factor (VEGF) and pro-angiogenic chemokines. TAMs are key players in immunosuppression in TME. The release of TGF-β, IL-10, indoleamine 2,3-dioxygenase (IDO), and prostaglandins encourages the expansion of regulatory T cells (Treg cells), an improper displacement of dendritic cells towards an immature and tolerogenic condition, and a deficit of T cells in metabolism. Immunosuppressive TAMs feature a high expression of immune checkpoint molecules (PD-L1, PD-L2, and B7-H4), which lead to T cell depletion. EMT = epithelial-mesenchymal transition; ILC3 = type-3 innate lymphoid cell; TH17 = T helper 17.

**Figure 10 biomolecules-14-00184-f010:**
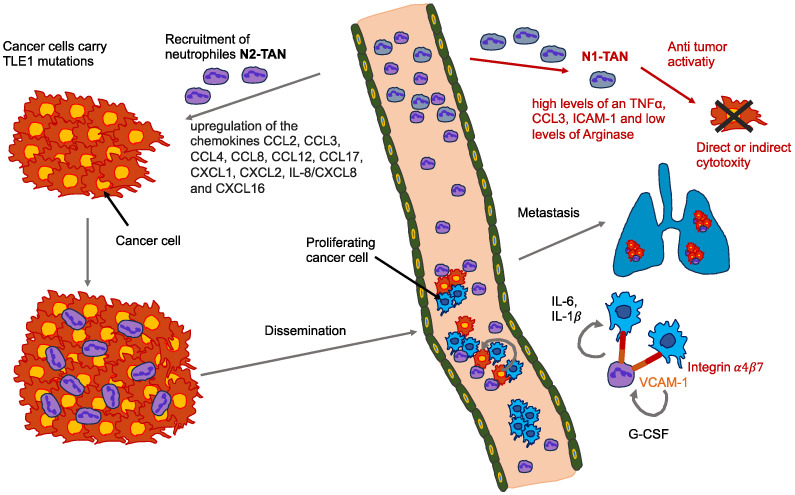
Bidirectionality of TANs: Cancer cells with mutations in distinct genes, such as TLE1, attract neutrophils toward the primary tumor region and elevate the number of CTC-neutrophil clusters within the bloodstream. Expression of the vascular cell adhesion molecule-1 (VCAM-1) in cancer cells fosters the assembly of CTC-neutrophil clusters, probably by tethering to distinct integrins on the cell surface of neutrophils. These neutrophils express IL-6 and IL-1β, which are cues that encourage CTC’s cell-cycle progression. In contrast, CTCs express neutrophil-stimulating factors like G-CSF. CTC-neutrophil clusters are extremely effective in the initiation of metastasis, and their existence is connected with a negative prognosis.

**Figure 11 biomolecules-14-00184-f011:**
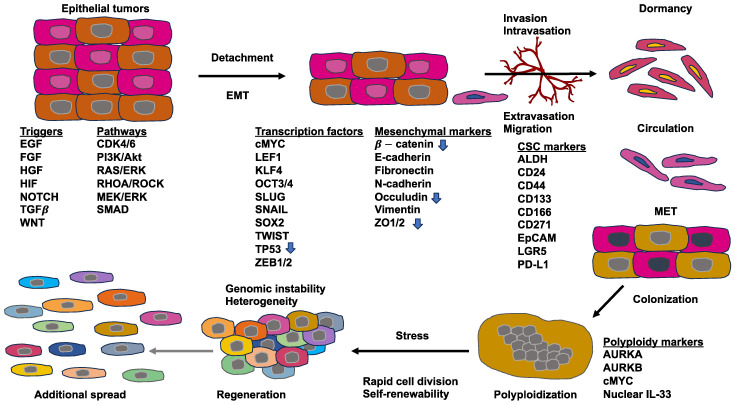
The primary cancer cells can undergo an epithelial-mesenchymal transition (EMT) and the reverse process, the mesenchymal-epithelial transition (MET). Thereby, different intermediate states are possible. A successful metastasis can also include polyploidization. The molecules designated denote representative examples that promote and foster the malignant progression of cancer. However, there are also downregulated molecules, which are indicated by an arrow and would impair the process of metastasis if they were not downregulated.

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
