# Peer review of "Phenotypic Heterogeneity, Bidirectionality, Universal Cues, Plasticity, Mechanics, and the Tumor Microenvironment Drive Cancer Metastasis"

_biomolecules, 2024, doi:10.3390/biom14020184_

Round 1
Reviewer 1 Report
Comments and Suggestions for Authors
The article represents a comprehensive review in the field of ‘Mechanobiology of cancer’ and discusses, generally, the concept of a “mechanical phenotype” of a tumor cells and factors contributing its development and diversification. On the one hand, the author describes universal mechanical properties required for the in situ establishment of a primary tumor and mechanical properties under the fluid flow (in case of extravasation and metastasis). On the other hand, biomechanical characteristics of specifically tumor cells and the surrounding components (stromal cells, ECM, vessels) are illuminated. And, as I can understand, at every step the author tries to demonstrate the link between mechanics and altered cancer genomics, epigenomics, metabolomics, biochemistry.
The paper is well-written, at a high language level. It may be interesting and helpful to a wide range of readers (including oncologists) who are not direct specialists in biomechanics.
The are some considerations probably worth of noting:
I wouldn’t generally recommend making the text shorter as shortening may decrease its comprehensiveness and, thus, understanding of its conception (indeed, at a first glance such a suggestion may arise – 29 pages of text, like a book chapter, but ‘too long’ may be only a first impression). Nevertheless, there are some points in the text where, to my opinion, it would be better to be more concise focusing directly on biomechanical aspects (especially in view of the journal series “Cell Mechanics in Cell Biology and Biological Matter Physics”); otherwise ‒ include a clearer explanation of how this or that issue relates to the main topic. I mean the following points, for example:
(a) Paragraph 5 (describing the role of exosomes, ctDNA, and genomic instability) – how does it relate to the problem of a mechanical phenotype? What is the logical link with other paragraphs and the need to include a detailed description of these 3 phenomena?
(b) Paragraph 9 (especially its preamble) – too many general facts about pro- and antitumoral activity of TAMs which are the subject of a separate paper, and it seems better to concentrate on their mechanics-related impact (which is indeed very intriguing).
(c) Some purely immunological aspects in paragraph 8 (such as immune checkpoints and cytokines expression) seem not directly linked to the main problem and thus excessive in description. Probably it would be sufficient to emphasize on mechanics-based mechanisms of immune evasion (eg, “masking”, “coating”, etc).
And there are few technical notions:
Figure 1 caption: not “left/right” parts of the diagram, but upper/lower;
Figure 2 needs a short common title.
Figure 3 illustrates the specificity of biological meanings of the three laws (Poiseuille, Starling and Darcy’s) which together describe the flux within blood vessels and tissues. But the figure caption should be slightly extended because otherwise its meaning is not very clear (and how it is related to the 3 equations). Some more description is needed for the designations shown in this diagram.
Line 742: ‘netroptosis’ should be corrected to necroptosis (a typo error)
Author Response
Dear Reviewer 1
Thank you for your helpful and valuable comments. I have taken them all into account, which has made the manuscript somewhat longer. I also had to edit the comments of the second reviewer, which also contributed to making it longer. The manuscript is now much better focused and the paragraphs are linked to the mechanophenotype. All changes are highlighted in yellow.
The article represents a comprehensive review in the field of ‘Mechanobiology of cancer’ and discusses, generally, the concept of a “mechanical phenotype” of a tumor cells and factors contributing its development and diversification. On the one hand, the author describes universal mechanical properties required for the in situ establishment of a primary tumor and mechanical properties under the fluid flow (in case of extravasation and metastasis). On the other hand, biomechanical characteristics of specifically tumor cells and the surrounding components (stromal cells, ECM, vessels) are illuminated. And, as I can understand, at every step the author tries to demonstrate the link between mechanics and altered cancer genomics, epigenomics, metabolomics, biochemistry.
The paper is well-written, at a high language level. It may be interesting and helpful to a wide range of readers (including oncologists) who are not direct specialists in biomechanics.
The are some considerations probably worth of noting:
I wouldn’t generally recommend making the text shorter as shortening may decrease its comprehensiveness and, thus, understanding of its conception (indeed, at a first glance such a suggestion may arise – 29 pages of text, like a book chapter, but ‘too long’ may be only a first impression). Nevertheless, there are some points in the text where, to my opinion, it would be better to be more concise focusing directly on biomechanical aspects (especially in view of the journal series “Cell Mechanics in Cell Biology and Biological Matter Physics”); otherwise ‒ include a clearer explanation of how this or that issue relates to the main topic. I mean the following points, for example:
(a) Paragraph 5 (describing the role of exosomes, ctDNA, and genomic instability) – how does it relate to the problem of a mechanical phenotype? What is the logical link with other paragraphs and the need to include a detailed description of these 3 phenomena?
Answer: I agree and rewrote the manuscript in paragraph 5. It is now better explained and connected to the mechanical phenotype. The linkage to others parts is now clear.
(b) Paragraph 9 (especially its preamble) – too many general facts about pro- and antitumoral activity of TAMs which are the subject of a separate paper, and it seems better to concentrate on their mechanics-related impact (which is indeed very intriguing).
Answer: I agree and have rewritten paragraph 9 by linking it to the topic of mechanics. Since I am also addressing readers from other disciplines with my review article, these basic parts will help them to understand the manuscript.
(c) Some purely immunological aspects in paragraph 8 (such as immune checkpoints and cytokines expression) seem not directly linked to the main problem and thus excessive in description. Probably it would be sufficient to emphasize on mechanics-based mechanisms of immune evasion (eg, “masking”, “coating”, etc).
Answer: I understand the criticism here and have now made it clear in the manuscript how these aspects influence these factors with the mechanical changes. Now it is comprehensible for the reader. I hope it meets with your approval.
And there are few technical notions:
Figure 1 caption: not “left/right” parts of the diagram, but upper/lower;
Answer: Thank you. Done.
Figure 2 needs a short common title.
Answer: I agree. Done.
Figure 3 illustrates the specificity of biological meanings of the three laws (Poiseuille, Starling and Darcy’s) which together describe the flux within blood vessels and tissues. But the figure caption should be slightly extended because otherwise its meaning is not very clear (and how it is related to the 3 equations). Some more description is needed for the designations shown in this diagram.
Answer: I totally agree and extended the figure legends and provided a new figure caption.
Line 742: ‘netroptosis’ should be corrected to necroptosis (a typo error)
Answer: Thank you. It is corrected.
Best regards
Claudia Tanja Mierke

Reviewer 2 Report
Comments and Suggestions for Authors
Tumor diseases become a huge problem when they embark on a pathway that leads to an increase in malignancy, such as the process of metastasis. Apart from the biochemical characteristics, tumor treatments also rely on the tumor microenvironment, which is recognized to be immunosuppressive and, as has recently been found, mechanically stimulates cancer cells and thus alters their functions. This review article highlights the interaction of cancer cells with other cells in the vascular metastatic route and discusses the impact of this intercellular interplay on the mechanical characteristics and subsequently on the behavior of cancer cells. The topic is interesting and important. Moreover, the manuscript is well organized and logically structured. However, there are issues to be considered.
Major concern
l Please discuss more about how stiff matrix regulates cancer stemness. What signal pathways or transcriptional factors are activated.
l Please discuss whether stiff matrix induces exosome secretion and protein (on the membrane or in the lumen of the exosome) or miRNA expressions in the exosomes from cancer cells to promote tumor growth.
Minor
l Line 236. Please give the full name of “AJs”.
l Line 419. “mimic circulating tumor cells (m-CTCs)” is already defined at line 411. I suggest to revise as “…and referred to as m-CTS”.
l Line 701. Please give the full name of “TGF-b”. Line 787 and 879, please give abbreviation of “transforming growth factor-β”.
l Line 704. “cancer stem cells (CSCs)” is already defined at line 121. I suggest to revise as “…of liver CSCs…”.
Author Response
Dear Reviewer 2
Thank you for your valuable comments that I have all addressed and included in my review article. The changes are highlighted in yellow.
Tumor diseases become a huge problem when they embark on a pathway that leads to an increase in malignancy, such as the process of metastasis. Apart from the biochemical characteristics, tumor treatments also rely on the tumor microenvironment, which is recognized to be immunosuppressive and, as has recently been found, mechanically stimulates cancer cells and thus alters their functions. This review article highlights the interaction of cancer cells with other cells in the vascular metastatic route and discusses the impact of this intercellular interplay on the mechanical characteristics and subsequently on the behavior of cancer cells. The topic is interesting and important. Moreover, the manuscript is well organized and logically structured. However, there are issues to be considered.
Major concern
l Please discuss more about how stiff matrix regulates cancer stemness. What signal pathways or transcriptional factors are activated.
Answer: I fully agree and included a discussion in this in the manuscript.
l Please discuss whether stiff matrix induces exosome secretion and protein (on the membrane or in the lumen of the exosome) or miRNA expressions in the exosomes from cancer cells to promote tumor growth.
Answer: I fully agree and included a discussion in this important issue in the manuscript.
Minor
l Line 236. Please give the full name of “AJs”.
Answer: Thank you. It is now provided.
l Line 419. “mimic circulating tumor cells (m-CTCs)” is already defined at line 411. I suggest to revise as “…and referred to as m-CTS”.
Answer: Thank you. Done.
l Line 701. Please give the full name of “TGF-b”. Line 787 and 879, please give abbreviation of “transforming growth factor-β”.
Answer: Thank you. Done.
l Line 704. “cancer stem cells (CSCs)” is already defined at line 121. I suggest to revise as “…of liver CSCs…”.
Answer: Thank you. Done.
Best regards
Claudia Tanja Mierke
